# Measurement of the superfluid fraction of a supersolid by Josephson effect

G. Biagioni[1,2,9], N. Antolini[2,3,9], B. Donelli[3,4,5,6], L. Pezzè[3,4,5], A. Smerzi[3,4,5 ✉], M. Fattori[1,3,7], A. Fioretti[2], C. Gabbanini[2], M. Inguscio[3,8], L. Tanzi[2,3] & G. Modugno[1,2,3 ✉]

A new class of superfluids and superconductors with spatially periodic modulation of the superfluid density is arising[1–12]. It might be related to the supersolid phase of matter, in which the spontaneous breaking of gauge and translational symmetries leads to a spatially modulated macroscopic wavefunction[13–16]. This relation was recognized only in some cases[1,2,5–9] and there is the need for a universal property quantifying the differences between supersolids and ordinary matter, such as the superfluid fraction, which measures the reduction in superfluid stiffness resulting from the spatial modulation[16–18]. The superfluid fraction was introduced long ago[16], but it has not yet been assessed experimentally. Here we demonstrate an innovative method to measure the superfluid fraction based on the Josephson effect, a ubiquitous phenomenon associated with the presence of a physical barrier between two superfluids or superconductors[19], which might also be expected for supersolids[20], owing to the spatial modulation. We demonstrate that individual cells of a supersolid can sustain Josephson oscillations and we show that, from the current–phase dynamics, we can derive directly the superfluid fraction. Our study of a cold-atom dipolar supersolid[7] reveals a relatively large sub-unity superfluid fraction that makes realistic the study of previously unknown phenomena such as partially quantized vortices and supercurrents[16–18]. Our results open a new direction of research that may unify the description of all supersolid-like systems.

Supersolids are a fundamental phase of matter originated by the spontaneous breaking of the gauge symmetry as in superfluids and superconductors and of the translational symmetry as in crystals[13–16]. This gives rise to a macroscopic wavefunction with spatially periodic modulation and to mixed superfluid and crystalline properties. Supersolids were originally predicted in the context of solid helium[13–16]. Today, quantum phases with spontaneous modulation of the wavefunction are under study in a variety of bosonic and fermionic systems. These include: the second layer of $^4$He on graphite[1,2]; ultracold quantum gases in optical cavities[5], with spin–orbit coupling[6] or with strong dipolar interactions[7–9,21]; the pair-density-wave phase of $^3$He under confinement[3,4]; and pair-density-wave phases in various types of superconductor[10–12]. Related phases have been observed in frustrated magnetic systems[22] or proposed to exist in the crust of neutron stars[23] and for excitons in semiconductor heterostructures[24]. The periodic structure of the wavefunction of all these systems is a prerequisite for supersolidity, which has so far, however, emerged clearly only in some cold-atom systems with the evidence of the double spontaneous symmetry breaking and of the mixed superfluid-crystalline character[5,25,26]. The experiments carried out so far on the other types of system have proved the coexistence of superfluidity/superconductivity and crystal-like structure[1–4,10–12], but no quantitative connection of the observations to the concept of supersolidity has been made. One of the difficulties

in comparing different types of system with spatial modulation of the wavefunction is the seeming lack of a universal property quantifying the deviations from the dynamical behaviour of ordinary superfluids or superconductors.

Here we note that a property with such characteristics already exists, the so-called superfluid fraction of supersolids, well known in the field of superfluids but not in that of superconductors. The superfluid fraction, introduced by A. J. Leggett in 1970 (ref. 16), quantifies the effect of the spatial modulation on the superfluid stiffness, which is in itself a defining property of superfluids and superconductors. The superfluid stiffness indeed measures the finite energy cost of twisting the phase of the macroscopic wavefunction and accounts for all fundamental phenomena of superfluidity, such as phase coherence, quantized vortices and supercurrents[27]. As sketched in Fig. 1, whereas in a homogeneous superfluid/superconductor the phase varies linearly in space, in a modulated system, most of the phase variation can be accommodated in the minima of the density, reducing the energy cost. Because the superfluid velocity is the gradient of the phase, this implies that peaks and valleys should move differently, giving rise to complex dynamics with mixed classical (crystalline) and quantum (superfluid) character. For example, fundamental superfluid phenomena such as vortices and supercurrents are predicted to be profoundly affected by the presence of the spatial modulation, losing the canonical quantization of their

[1]Dipartimento di Fisica e Astronomia, Università degli studi di Firenze, Sesto Fiorentino, Italy. [2]CNR-INO, Sede di Pisa, Pisa, Italy. [3]European Laboratory for Non-Linear Spectroscopy, Università degli studi di Firenze, Sesto Fiorentino, Italy. [4]CNR-INO, Sede di Firenze, Firenze, Italy. [5]Quantum Science and Technology in Arcetri (QSTAR), Firenze, Italy. [6]Università degli Studi di Napoli Federico II, Napoli, Italy. [7]CNR-INO, Sede di Sesto Fiorentino, Sesto Fiorentino, Italy. [8]Dipartimento di Ingegneria, Università Campus Bio-Medico di Roma, Roma, Italy. [9]These authors contributed equally: G. Biagioni, N. Antolini. ✉e-mail: augusto.smerzi@cnr.ino.it; modugno@lens.unifi.it

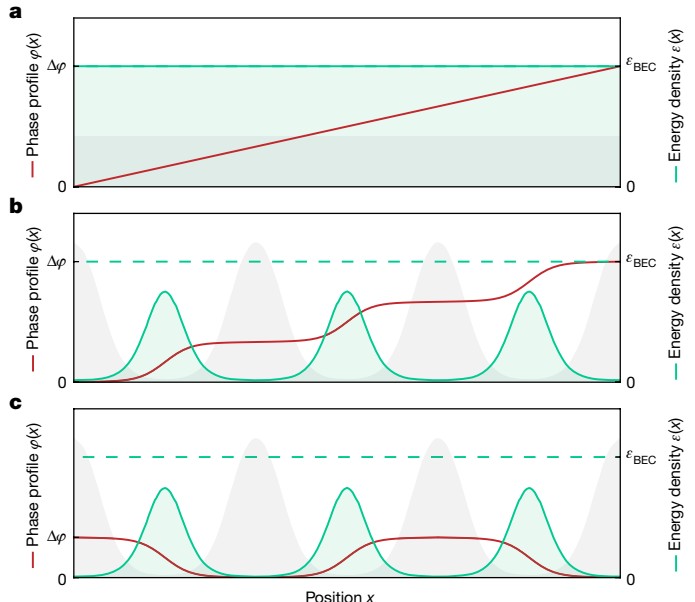

**Fig. 1 | Superfluid fraction in superfluids and supersolids.** Sketches of the superfluid fraction from the application of a phase twist in a bosonic system at zero temperature. **a**, In a homogeneous superfluid, a phase twist with amplitude $\Delta\varphi$ results in a constant gradient of the phase, that is, a constant velocity, whereas in a supersolid (**b**,**c**), the kinetic energy can be minimized by accumulating most of the phase variation in the low-density regions. The grey and green areas represent the number density and the kinetic energy density, respectively, whereas the phase profile is plotted in red. The superfluid fraction is the ratio of the area under the green curve to that of the homogeneous case. **b**, Leggett's approach, which—for an annular system—would correspond to a stationary rotation, leads to a monotonous increase of the phase. **c**, Our method, based on an alternating oscillation of the phase, leads to Josephson oscillations. Both kinetic energy and superfluid fraction are the same for **b** and **c**.

angular momentum[16–18,28]. The superfluid fraction, which ranges from unity for standard superfluids to zero for standard crystals, enters directly in all these phenomena and is therefore the proper quantity to assess the deviations from standard superfluids and superconductors. Note that the superfluid fraction of supersolids is not related to thermal effects, in contrast to the superfluid fraction owing to the thermal depletion of superfluids and superconductors[29].

The standard methods to measure the superfluid stiffness are based on the measurement of global properties such as the moment of inertia for rotating superfluids[1,2] or the penetration depth of the magnetic field for superconductors[30]. In dipolar supersolids, previous attempts using rotational techniques revealed a large superfluid fraction[31] but were not precise enough to assess its sub-unity value[32,33]. In the other systems, there is evidence that the superfluid stiffness is low[1,2,30], but no quantitative measurement of a sub-unity superfluid fraction is available.

In this work, we demonstrate that it is possible to measure the superfluid fraction of a supersolid not only from global dynamics but also from a fundamental phenomenon taking place in individual cells of the supersolid lattice: the Josephson effect[19]. As sketched in Fig. 1, the unit cell of a 1D supersolid lattice is composed by two density maxima connecting through a density minimum, so it has the typical structure of a Josephson junction, two bulk superfluids connected by a weak link. It is therefore tempting to associate supersolidity to the very existence of local Josephson dynamics. So far, the analogy between a supersolid and an array of Josephson junctions has only been used to model the relaxation towards the ground state of a dipolar supersolid[20]. There is instead no theoretical or experimental evidence for local Josephson oscillations or an understanding of the potential relation between the Josephson effect and the superfluid fraction. The problem is complicated by the

fact that, in supersolids, the weak links are self-induced by internal interactions rather than by an external potential, so they can change during the dynamics. Therefore, it is not clear if phenomena such as Josephson oscillations can exist at all in a supersolid.

Here we demonstrate experimentally and theoretically that a supersolid can, in fact, sustain coherent phase-density oscillations, behaving as an array of Josephson junctions. We also show that the Josephson coupling energy that we can deduct from the Josephson oscillations provides a direct measurement of the superfluid fraction. We use this new approach to measure with high precision the superfluid fraction of the dipolar supersolid appearing in a quantum gas of magnetic atoms. We find a range of sub-unity values of the superfluid fraction, depending on the depth of the density modulation in accordance with Leggett's predictions.

Leggett's approach to the superfluid fraction considers an annular supersolid in the rotating frame and maps it to a linear system with an overall phase twist, as sketched in Fig. 1b. The superfluid fraction is defined on a unit cell as[16,34]

$$f_s = \frac{E_{kin}}{E_{kin}^{hom}}. \tag{1}$$

The numerator is the kinetic energy acquired by the supersolid with number density $n(x)$ when applying a phase twist $\Delta\varphi$ over a lattice cell of length $d$, $E_{kin} = \hbar^2/(2m) \int_{cell} dx\, n(x)\, |\nabla\varphi(x)|^2$, and thus accounts for density and phase modulations. The denominator $E_{kin}^{hom} = Nmv_s^2/2$ is the kinetic energy of a homogeneous superfluid of $N$ atoms and velocity $v_s = \hbar\Delta\varphi/(md)$ associated with a constant phase gradient $\Delta\varphi$ across the cell. Using a variational approach[16,35], Leggett found an upper and a lower bound for equation (1), $f_s^l \le f_s \le f_s^u$; see Methods. In particular, the upper bound

$$f_s^u = \left(\frac{1}{d} \int_0^d \frac{dx}{\bar{n}(x)}\right)^{-1}, \tag{2}$$

in which $\bar{n}(x)$ is the normalized 1D density, restricts $f_s$ to be lower than unity if the density is spatially modulated. Note that the calculation of the superfluid fraction, which is a global property, by considering a single lattice cell is possible owing to the periodicity of the wavefunction of the supersolid[16].

We propose an alternative expression for the superfluid fraction, considering Josephson phase twists with alternating sign between neighbouring lattice sites of a supersolid, as sketched in Fig. 1c. This corresponds to a different type of motion of the supersolid, with no global flow but with alternate Josephson phase-density oscillations between sites. Also, in this case, we can consider a single cell, because the kinetic energy is proportional to $|\nabla\varphi(x)|^2$, so it does not depend on the sign of the phase twist. In the limit of small excitations ($\Delta\varphi \to 0$), the kinetic energy of a Josephson junction is given by $E_{kin} = NK\Delta\varphi^2$, in which $K$ is the coupling energy across the barrier[36]. From equation (1), we thus find:

$$f_s = \frac{K}{\hbar^2/(2md^2)}, \tag{3}$$

showing a direct relation between the superfluid fraction and the coupling energy of the junction. We note that an expression similar to the upper bound in equation (2) was derived by Leggett for the coupling energy of a single Josephson junction[37], however without discussing the connection to the superfluid fraction.

We now demonstrate the existence of coherent Josephson-like oscillations in a dipolar supersolid[7–9]. This system is particularly appealing to study fundamental aspects of supersolidity[38]: the supersolid lattice is macroscopic, with many atoms per site and large superfluid effects; the available control of the quantum phase transition allows to directly

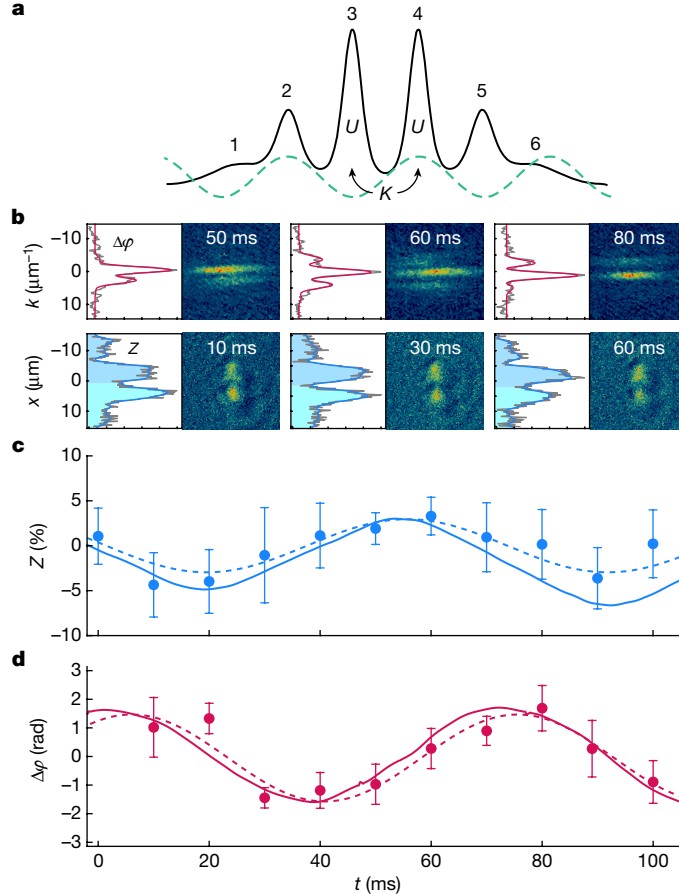

**Fig. 2 | Josephson oscillations in a supersolid. a**, Sketch of the experimental system. The black line is the supersolid density profile at equilibrium. The dashed green line is the optical lattice potential used for the phase imprinting. **b**, Examples of experimental single shots and corresponding integrated 1D profiles. Top row, interference fringes after a free expansion. Red curves are fit functions used to extract the phase difference $\Delta\varphi$. Bottom row, in situ images. Shaded areas indicate the populations of the left and right halves of the supersolid used to extract the population imbalance $Z$. **c**, Oscillations of $Z$ as a function of time at $\varepsilon_{dd} = 1.428$. Dots are experimental points. Error bars are the s.e.m. of 20–30 measurements. The solid line is the numerical simulation for the same parameters. The dashed line is a sinusoidal fit to the experimental data. **d**, Same for $\Delta\varphi$. Experimental values and error bars are calculated using the circular mean and s.e.m. (see Methods).

compare supersolids and superfluids; and interactions are weak, allowing a fairly accurate theoretical modelling[39]. Our experimental system[7] is composed of about $N = 3 \times 10^4$ bosonic dysprosium atoms, held in a harmonic trap elongated along the $x$ direction, with trap frequencies $(\omega_x, \omega_y, \omega_z) = 2\pi(18, 97, 102)$ Hz. By tuning the relative strength $\varepsilon_{dd}$ of dipolar and contact interactions, we can cross the quantum phase transition from a standard Bose–Einstein condensate (BEC) to the supersolid regime (Methods). The supersolid lattice structure is 1D, leading to a continuous phase transition[40]. Our typical supersolid is made of two main central clusters and four smaller lateral ones, with a lattice period $d \simeq 4$ μm, as shown in Fig. 2. We can vary the density modulation depth by varying the interaction strength in the range $\varepsilon_{dd} = 1.38$–1.45; further increasing $\varepsilon_{dd}$ leads to the formation of an incoherent crystal of separate clusters, the so-called droplet crystal, a regime that we cannot study experimentally because of its short lifetime[7].

Because our system is inhomogeneous, we focus our attention on the central cell, the one delimited by clusters 3 and 4 in Fig. 2. As we will show, the superfluid fraction we derive from that cell corresponds to the superfluid fraction of a hypothetical homogeneous supersolid

with all cells identical to the central cell, as in Fig. 1, which is the system of general interest.

We find that the application for a short time of an optical lattice with twice the spacing of the supersolid (sketched in Fig. 2a) imprints the proper alternating phase difference between adjacent clusters to excite Josephson oscillations. With a depth of 100 nK and an application time of $\tau = 100$ μs, we obtain a phase difference on the order of π/2. After a variable evolution time in the absence of the lattice, we measure both the evolving phase difference $\Delta\varphi$ between neighbouring clusters and the population difference $Z$ between the left and right halves of the supersolid. $\Delta\varphi$ is measured from the interference fringes developing after a free expansion (snapshots in Fig. 2b, top row), whereas $Z$ is measured by in situ phase-contrast imaging (Fig. 2b, bottom row) (Methods). As shown in Fig. 2c,d, we observe single-frequency oscillations of $Z$ and $\Delta\varphi$, with the characteristic π/2 phase shift of the standard Josephson dynamics[19,36,41–44]. The observation time is limited to about 100 ms by the finite lifetime of the supersolid, owing to unavoidable particle losses[7]. The experimental observations agree very well with numerical simulations based on the time-dependent extended Gross–Pitaevskii equation (GPE), also shown in Fig. 2c,d (Methods). We have checked that the Josephson oscillations are not observable if we apply the same procedure to standard BECs instead of supersolids (see Methods).

The observation of a single frequency in both experiment and simulations indicates that not only is it possible to excite Josephson-like oscillations in a supersolid but also they are a normal mode of the system. To model our observations, we develop a six-mode model, generalizing the two-mode Josephson oscillations[36] to the case of six clusters (see Methods). We associate to the $j$th cluster a population $N_j$ and a phase $\varphi_j$ ($j = 1,...,6$). In general, the dynamics includes contributions from each cluster and shows several frequencies. However, we find that, under appropriate conditions among the interaction and coupling energies, there exists a normal mode of the system in which the dynamical variables of the two central clusters of the supersolid decouple from the lateral ones. This results in Josephson-like oscillations described by the two coupled equations

$$\Delta\dot{N} = -4KN_{34}\sin(\Delta\varphi) \tag{4a}$$

$$\Delta\dot{\varphi} = U\Delta N \tag{4b}$$

in which $\Delta N = N_3 - N_4$, $N_{34} = N_3(0) + N_4(0)$, $\Delta\varphi = \varphi_3 - \varphi_4$ and $U$ is the interaction energy per particle. These equations hold for interaction energies $N_{34}U$ much larger than $K$ (for our system, $N_{34}U/(2K) > 25$; see Methods). Because in our case $\Delta N \ll N$, we keep only linear terms in $\Delta N/N$.

Equations (4a) and (4b) are equivalent to those of a simple pendulum with angle $\Delta\varphi$ and angular momentum $\Delta N$ and, in the small-angle limit, feature sinusoidal oscillations with a single frequency, $\omega_J = \sqrt{4KUN_{34}}$. We emphasize that the current–phase relation equation (4a) as well as $\omega_J^2$ differ by a factor of 2 with respect to the Josephson equations of two weakly coupled BECs, owing to the contribution of the lateral clusters, but are equal to those of a hypothetical homogeneous supersolid. Notice also that equations (4a) and (4b) depend only on the coupling energy $K$ and the interaction energy $U$ of the two central clusters, in contrast to the expectation that the inhomogeneity of the trapped system may introduce other energies in the equations of motion. We checked by Gross–Pitaevskii simulations that our experimental configuration satisfies the conditions to have a Josephson-like normal mode (namely, equation (7) in Methods).

In the experiment, we are not able to resolve the population of the individual clusters but we study the population difference between the left and right halves of the system, $Z = (N_1 + N_2 + N_3 - N_4 - N_5 - N_6)/N$. There is a proportionality relation between the two observables, $\Delta N = 2NZ$, which allows us to rewrite equations (4a) and (4b) in terms of the experimental observables (Methods).

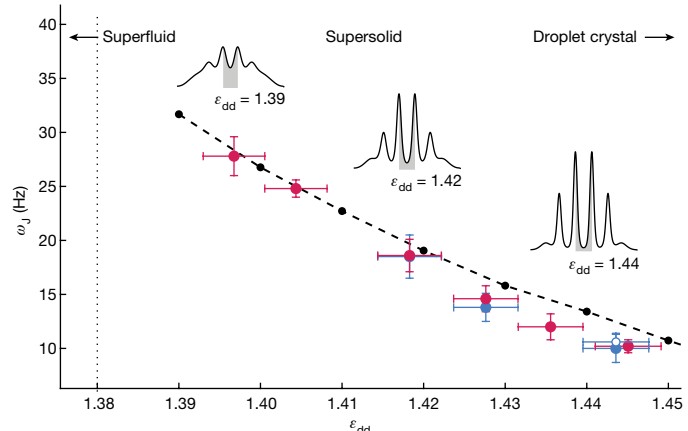

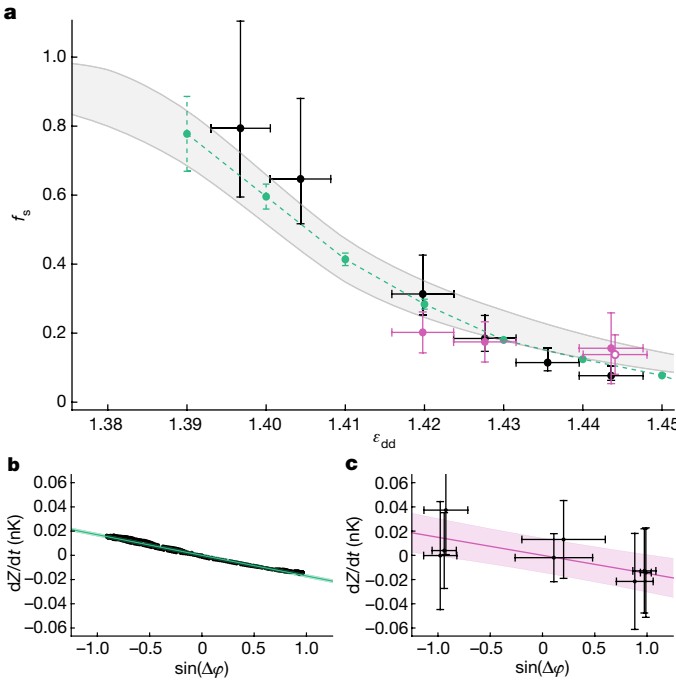

**Fig. 3 | Josephson oscillation frequency versus the interaction parameter.** Red dots are the experimental frequencies for $\Delta\varphi$. Filled and open blue dots are the frequencies for $Z$ measured by in situ imaging with and without optical separation, respectively (Methods). Vertical error bars are the uncertainties in the nonlinear fit of the sinusoidal oscillations shown in Fig. 2c,d. Horizontal error bars represent the experimental resolution in $\varepsilon_{dd}$ (Methods). The red point at $\varepsilon_{dd} = 1.444$ is shifted slightly horizontally for clarity. Black points are the results of numerical simulations. The dashed line is a guide for the eye. The insets show the modulated ground-state density profiles obtained from numerical simulations for different values of $\varepsilon_{dd}$. The vertical dotted line marks the critical point of the superfluid–supersolid quantum phase transition.

An important difference between a cell of the supersolid and a standard Josephson junction is the fact that, in the supersolid, the position of the weak link is not fixed by an external barrier but it is self-induced, so it can move. This leads to the appearance of a low-energy Goldstone mode associated with the spontaneous translational symmetry breaking. In a harmonic potential, it consists of a slow oscillation of the position of the weak link, together with the density maxima, and an associated oscillation of both $Z$ and $\Delta\varphi$ (ref. 26). Owing to its low frequency (on the order of a few Hz), the Goldstone mode is spontaneously excited by thermal fluctuations, resulting in shot-to-shot fluctuations of the experimental observables. The same low frequency, however, allows to separate Josephson and Goldstone dynamics in both experiment and theory (Methods).

We measure the Josephson frequency $\omega_J$ from a sinusoidal fit of the phase and population dynamics in Fig. 2c,d. We repeat the measurement by varying the interaction parameter $\varepsilon_{dd}$, corresponding to different depths of the supersolid density modulation. Figure 3 shows the fitted frequencies as a function of $\varepsilon_{dd}$ and a comparison with the numerical simulations. We observe a decrease of the frequency for increasing $\varepsilon_{dd}$. This is justified by the fact that the superfluid current across the junction decreases because a larger and larger portion of the wavefunction remains localized inside the clusters (see insets in Fig. 3). This reduces the coupling energy $K$ while only weakly affecting the interaction energy.

From the Josephson frequency, we can derive the coupling energy as $K = \omega_J^2/(4UN_{34})$, with the denominator obtained from the simulations. We verified that this relation holds not only in the small-amplitude regime of the simulations but also for the larger amplitudes of the experiment.

From the measured $K$, we derive in turn the superfluid fraction using equation (3). The results are shown in Fig. 4 and feature a progressive reduction of the superfluid fraction below unity for increasing depths of the supersolid modulation. The experimental data are in good agreement with the numerical simulations (green dots), in which—according to equation (4a)—the coupling energy is obtained from the linear dependence of $dZ/dt$ on $\sin(\Delta\varphi)$ (current–phase relation); see Fig. 4b. A similar analysis (Fig. 4c) was performed on the experimental data for

**Fig. 4 | Superfluid fraction from Josephson oscillations. a**, Superfluid fraction as a function of $\varepsilon_{dd}$. Black dots are experimental results derived from the Josephson frequencies. Vertical error bars result from the error propagation of equation (3), with $K = \omega_J^2/(4UN_{34})$; see Methods. Green dots are results from numerical simulations. Error bars are the uncertainties of the linear fits used to determine $K$ and $UN_{34}$. Pink points are derived from the experimental phase–current relation, as in **c**. Error bars are estimated using the propagation of equation (3), with $K$ and its relative uncertainty extracted from linear fits of experimental data. The open pink point at $\varepsilon_{dd} = 1.444$ is the dataset without the optical-separation technique (Methods). The grey band extends between the upper and lower bounds of equation (1). **b,c**, Phase–current relation at $\varepsilon_{dd} = 1.444$. The points show the results of numerical simulations (**b**) and experimental measurements (**c**). From the linear regressions (green and pink lines), we extract the coupling energy $K$ according to equation (4a). Shaded regions are the confidence bands for one s.d.

which we have combined phase and population oscillations (pink dots in Fig. 4a). The results for these data points demonstrate the reduced superfluid fraction of the supersolid with no numerical input on the interaction energy $U$.

In Fig. 4a, we also compare our results with Leggett's prediction of equation (2), relating the superfluid fraction to the density modulation of the supersolid. From the numerical density profiles, we calculate both the upper bound $f_s^u$ and the corresponding lower bound[35] $f_s^l$, which delimit the grey area in Fig. 4a (see Methods). The two bounds would coincide if the density distribution were separable in the transverse coordinates $y$ and $z$. Because our supersolid lattice is 1D, the two bounds are close to each other. The superfluid fraction calculated from the simulated dynamics lies between the two bounds in the whole supersolid region we investigated, demonstrating the applicability of Leggett's result to our system.

In conclusion, the overall agreement between experiment, simulations and theory on our dipolar supersolid proves the long-sought sub-unity superfluid fraction of supersolids and its relation to the spatial modulation of the superfluid density. The demonstration of self-sustained Josephson oscillations in a supersolid provides a new proof of the extraordinary nature of supersolids compared with ordinary superfluids and crystals. These oscillations indeed cannot exist neither in crystals, in which particles are bound to lattice sites, nor in ordinary superfluids, which do not have a lattice structure.

Our findings open new research directions. The observed reduction of the superfluid fraction with increasing modulation depths may explain the low superfluid stiffness measured in other systems, such as [4]He on graphite[1,2] or superconductors hosting pair-density-wave phases[10–12]. An important question related to the pair-density waves in fermionic systems is how Leggett's bounds on the superfluid fraction may be extended to systems in which the superfluid density and particle density do not coincide. Note that equation (2) is also applicable to standard superfluids with an externally imposed spatial modulation, as demonstrated for BECs in optical lattices by means of measurements of the effective mass[41] or of the sound velocity[45,46]. In the supersolid, however, the dynamics linked to the reduced superfluid fraction is not constrained by an external potential and so totally new phenomena might be observed. The large value of $f_s$ we measured for the dipolar supersolid, which remains larger than 10% also for deep density modulations, indicates that partially quantized supercurrents[16,18] and vortices[17] should appear at a macroscopic level.

Owing to the generality of the Josephson effect, our Josephson-oscillation technique might be applied to characterize the local superfluid dynamics of the other supersolid-like phases under study in superfluid and superconducting systems. Equation (3) is applicable in general, considering that the detection of Josephson oscillations implies measurement of both the coupling energy and the spatial period of the superfluid density modulation. For example, a promising type of system may be the pair-density-wave phase in superconductors, in which the modulation has already been resolved. The Josephson-oscillation technique works naturally in linear geometries and so it does not require any adaptation for the finite size of the clusters in the supersolid-like phases available in experiments[1–12], differently from the rotation technique[16] (see Methods).

Furthermore, the self-induced Josephson junctions we have identified in supersolids might have extraordinary properties resulting from the mobility of the weak links. Indeed, although the Goldstone mode of the weak links is not relevant for the Josephson dynamics owing to its very low energy, for the same reason, it may affect the fluctuation properties of the junction[47], potentially leading to new thermometry methods[48] and especially to previously unknown entanglement properties[49].

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

## Methods

### Supersolid preparation

The experiment starts from a BEC of $^{162}$Dy atoms trapped in a harmonic potential created by two dipole traps crossing in the horizontal $(x, y)$ plane[50]. To tune the interaction parameter $\varepsilon_{dd} = a_{dd}/a_s$, we control the contact scattering length $a_s$ with magnetic Feshbach resonances, whereas the dipolar scattering length $a_{dd} = 130a_0$ is fixed. The condensate is initially prepared at a magnetic field $B \simeq 5.5$ G, corresponding to a scattering length of about $140a_0$. The magnetic field is then slowly changed towards the critical values for the superfluid–supersolid phase transition, close to the set of Feshbach resonances around 5.1 G (refs. 7,39). We calibrate the magnetic-field amplitude using radio-frequency spectroscopy before and after each experimental run. The magnetic-field stability is about 0.5 mG, corresponding to a scattering-length stability of about $0.25a_0$. Because the overall systematic uncertainty in the absolute value of $a_s$ is about $3a_0$, corresponding to an uncertainty in $\varepsilon_{dd}$ of about 4%, we identify a precise $B$-to-$a_s$ conversion by comparing the experimental and numerical data for the critical $\varepsilon_{dd}$ for the phase transition[40]. The typical atom number in the supersolid is $N = (2.8 \pm 0.3) \times 10^4$. We expect thermal effects to be negligible in the Josephson dynamics, as the coupling energy $K(N_2 + N_3)$ is on the order of $k_B 100$ nK in the whole supersolid regime and from measurements of the thermal fraction on the BEC side, we get $T < 30$ nK (ref. 7).

### Excitation of the Josephson dynamics

The optical lattice used to excite the Josephson dynamics consists of two repulsive laser beams at 1,064 nm that intersect at a small angle, providing a lattice period $d_L = (7.9 \pm 0.3)$ μm. The stability of the lattice position is better than 10% of its period over the duration of the experiment (see Extended Data Fig. 1a) and, before each measurement, we check the relative position of the lattice and the supersolid. Most of the noise in the excitation protocol comes from shot-to-shot fluctuations of the supersolid lattice owing to the Goldstone mode (the position of a single cluster has a s.d. on the order of 25% of the supersolid period; see below).

To calibrate the phase difference imprinted by the optical lattice, we switch on the lattice at fixed $U_{lat} = k_B 100$ nK for a variable pulse duration $\tau$ and we measure the imprinted phase difference $\varphi_0 = U_{lat}\tau/\hbar$ immediately after the pulse; see Extended Data Fig. 1b.

In the experiment, we detect clear Josephson oscillations only when the initial imprinted phase is 1 rad or larger. In this regime, we compare the experimental and numerical Josephson frequencies as a function of the amplitude of the oscillation (see Extended Data Fig. 2). We find a small reduction (about 15%) compared with the small-amplitude regime, which allows us to use the equation $\omega_J = \sqrt{4KN_{34}U}$ to extract the coupling energy $K$ from the experimental Josephson frequencies. The need for large excitation amplitudes in the experiment can be explained by the presence of the Goldstone mode, which introduces an unavoidable noise on both $Z$ and $\Delta\varphi$.

We checked that applying the same phase-imprinting protocol to standard BECs does not produce any detectable Josephson oscillation; see Extended Data Fig. 3. This observation can be justified by the fact that the spatially stationary excitations of the condensate, the rotons, have a spatial period similar to the supersolid period $d$, so they cannot be excited by the optical lattice with $d_L \simeq 2d$. The excitations with spatial period equal to $d_L$ have instead a phonon/maxon character, they are not stationary in the harmonic trap and so they cannot produce spatially stable oscillations.

In general, this observation proves that the self-induced Josephson oscillations exist in the supersolid but not in the superfluid. We cannot make reliable experiments in the solid-like droplet-crystal phase, owing to the exceedingly short lifetime of the experimental system in that regime, but the simulations show that the Josephson coupling becomes negligible and Josephson oscillations are absent.

### Phase detection and analysis

To measure the phase difference between the two central clusters of the supersolid, $\Delta\varphi = \varphi_3 - \varphi_4$, we record the atomic distribution in the $(x, y)$ plane by absorption imaging after 61 ms of free expansion. About 200 μs before releasing the atoms from the trapping potential, we increase the contact interaction strength by setting $a_s = 140a_0$, thus minimizing the relative effects of the long-range dipolar interaction on the expansion. We interpret the recorded distributions as the atomic density in momentum space, $\rho(k_x, k_y)$. In the supersolid regime, the momentum distribution shows an interference pattern resulting from the superposition of the expanding matter waves of each cluster (see snapshots in Fig. 2a). We first integrate the 2D distribution over $k_y$ to obtain the 1D momentum distribution $\rho(k_x)$. We then fit $\rho(k_x)$ with a double-slit model:

$$\rho(k_x) = G(k_x, k_0, \sigma)[1 + A_1 \cos^2(\pi(k_x - k_0)/k_r + \theta)]$$

in which $G(k_x, k_0, \sigma)$ is a Gaussian envelope of centre $k_0$ and width $\sigma$ and $A_1$, $k_r$ and $\theta$ are the amplitude, period and phase of the modulation, respectively. Owing to the $\cos^2(x)$ term in our fit function, the physical phase difference is given by $\Delta\varphi = 2\theta$.

Although the interference pattern is generated by six overlapping clusters, $\Delta\varphi$ can be extracted with a good approximation (within 20%) by the double-slit model owing to the finite resolution of our imaging system in momentum space (0.2 μm$^{-1}$, 1/e Gaussian width) and to the lower weight of lateral clusters. This is experimentally confirmed by the measured imprinted phase $\varphi_0$ as a function of the pulse depth, shown in Extended Data Fig. 1, which is in good agreement with the prediction for the phase difference between adjacent clusters, $U_{lat}\tau/\hbar$.

For each observation time $t$, we take $n = 20$–$30$ images. We then calculate the mean value of $\Delta\varphi$ using the circular mean, which is appropriate for a periodic quantity such as an angle:

$$\overline{\Delta\varphi} = \arg\left(\sum_{j=1}^{n} e^{i\Delta\varphi_j}\right)$$

in which $\arg(x)$ indicates the argument of the complex number $x$ and i is the imaginary unit. The corresponding error is given by the circular s.d. of the mean[51].

### Imbalance detection and analysis

To measure the population imbalance between clusters, we image the supersolid in situ in the $(x, y)$ plane using an imaging system with a resolution of 2.5 μm, smaller than the cluster spacing of 4 μm. To avoid saturation effects as a result of the high density of the sample, we use dispersive phase-contrast imaging[52] with an optical beam detuned by $5\Gamma$ from the 421-nm optical transition. From each experimental shot, we calculate the imbalance as follows. We integrate the column density along the $y$ direction (transverse to the modulation), obtaining 1D density profiles in which we identify the two main peaks (snapshots in Fig. 2b). We then measure the populations $N_1 + N_2 + N_3$ and $N_4 + N_5 + N_6$ integrating the signal to the left and to the right of the minimum between the clusters, respectively. We then compute the observable $Z = (N_6 + N_5 + N_4 - N_3 - N_2 - N_1)/N$.

Owing to the limited optical resolution, we can only clearly resolve the left and right clusters populations when the contrast of the density modulation is high enough, that is, only at $\varepsilon_{dd} = 1.444$. For lower $\varepsilon_{dd}$, we use an optical-separation technique to increase the signal-to-noise ratio. We turn on the optical lattice used for the excitation 5 ms before image acquisition. This causes the main clusters to move away, falling into the minima of the optical potential and increasing their distance (snapshots in Fig. 2a and in Extended Data Fig. 4). Although our lattice does not have the optimal spatial phase to separate the clusters, because it has a maximum at the position of one cluster, we checked

with numerical simulations that the only effect on the imbalance $Z$ is the addition of a constant offset, thus not changing the oscillation frequency (see Extended Data Fig. 4). Experimentally, we checked that the Josephson frequencies measured with and without the optical separation are consistent within one s.d. (see filled and empty pink points at $\varepsilon_{dd} = 1.444$ in Fig. 4). At lower $\varepsilon_{dd}$, very close to the phase transition, the contrast is too low, so we rely only on phase measurements.

## Experimental measurement of the superfluid fraction from the Josephson frequency

To measure the superfluid fraction in the whole supersolid regime, reported in Fig. 4, we use the Josephson frequency $\omega_J$ extracted from phase oscillations. We use the formula $f_s = \dfrac{\omega_J^2/(4N_{34}U)}{\hbar^2/(2md^2)}$. The period $d$ of the supersolid lattice is measured with in situ imaging, obtaining $d = 3.7 \pm 0.1\,\mu m$. The quantity $N_{34}U$ is taken from the numerical simulations. Because the experimental oscillations are not in the small-amplitude limit, the frequencies are underestimated by about 15% (see Extended Data Fig. 2). The upper error bar for $f_s$ in Fig. 4 includes accordingly a 15% uncertainty. For the experimental configurations in which we measure both $Z$ and $\Delta\varphi$, we also checked that $U$ extracted from equation (5b) is in agreement with the simulations.

## Discussion of the Leggett model

Leggett derived the upper bound for the superfluid fraction $f_s^u$ in the case of a 1D system rotating in an annulus with radius $R$, for which the moment of inertia is $I = (1-f_s)I_c$, with $I_c$ the classical moment of inertia[16]. To find the phase profile $\varphi(x)$ that minimizes the kinetic energy for a fixed number density $n(x)$, we have to work in the frame corotating with the annulus, in which the external potential is independent of time. In this frame, the rotation imposes a phase twist between neighbouring clusters, proportional to the angular velocity $\Omega$ of the annulus, $\Delta\varphi = \varphi(d) - \varphi(0) = m\Omega Rd/\hbar$. The result of the energy minimization is $\varphi(x) = \Delta\varphi \int_0^x dx'n(x')^{-1}/\int_0^d dx'n(x')^{-1}$ and the corresponding kinetic energy cost is $E_{kin}(\Delta\varphi) = N\hbar^2/(2md^2)f_s^u \Delta\varphi^2$, in which $f_s^u$ is the upper bound of equation (3). The lower bound $f_s^l$, instead, is found starting from the 3D kinetic energy, which also includes the derivatives along the transverse directions $y$ and $z$ (ref. 34). It reads $f_s^l = \int dydz\,(1/d\int_0^d dx/\bar{n}(x,y,z))^{-1}$, in which $\bar{n}(x,y,z)$ is the normalized 3D density. From the expression of the energy, we see that the superfluid fraction has the role of an elastic constant for the phase deformation. The density and phase profiles sketched in Fig. 1b, and the corresponding energy density $\hbar^2/(2m)n(x)|\nabla\varphi(x)|^2$, are for a hypothetical homogeneous supersolid lattice with $f_s = 0.20$.

In the Josephson case, the phase twist is externally applied with an odd parity, to induce Josephson oscillations between neighbouring sites. The energy minimization on the single cell gives the same result as before, as it is insensitive to the sign of the phase twist. In the sketch of Fig. 1c, we build the odd phase profile by changing sign from cell to cell to $\varphi(x)$ of Fig. 1b. We note that, in a linear system such as that used in the experiment, the superfluid fraction measured from the Josephson dynamics is not affected by radial effects, which are instead relevant in the case of rotating systems[31]. Indeed, the superfluid fraction extracted from a measurement of the moment of inertia, $I = (1-f_s)I_c$, would also take into account the extra contribution given by the reduced inertia of the superfluid clusters composing the system, which rotate around their centres of mass. Leggett's upper bound is instead derived in the limit of an infinite radius of the annulus, for which such radial effects can be neglected[16].

## Goldstone mode

In a harmonic trap, the Goldstone mode energy $\hbar\omega_G$ is finite but much smaller than $\hbar\omega_x$, as the supersolid can rearrange its density to minimize the centre-of-mass displacement. The resulting dynamics is an oscillation of the lattice position, imbalance and relative phase. Owing to its low frequency, the Goldstone mode is thermally activated. Similarly to

previous works[26], we detect the Goldstone excitation as fluctuations in the lattice position that keep the centre of mass fixed (see Extended Data Fig. 5). We prepare the supersolid with three main clusters and, without any further manipulation, examine the in situ density. We detect fluctuations in the cluster positions, with s.d. $\sigma_{clusters} \approx 1\,\mu m$, much larger than the centre-of-mass fluctuations, $\sigma_{com} \approx 0.4\,\mu m$. The Goldstone mode also introduces some noise in $Z$ and $\Delta\varphi$ during the dynamics, which we estimate to be about 20% of the Josephson amplitude, for both observables.

The frequency of the Goldstone mode can be observed in numerical simulations at $T = 0$ by setting an initial $Z_0 > 0$ together with a small displacement of the weak link position, $x_0 \neq 0$; see Extended Data Fig. 5. In the time evolution of $Z$, we find a very low frequency oscillation, $\omega_G = 2\pi(3.56 \pm 0.08)$ Hz, on top of the Josephson dynamics, $\omega_J = 2\pi(23.85 \pm 0.03)$ Hz. We find similar values for $\Delta\varphi$. The weak link position oscillates at the same low frequency $\omega_G$.

## Numerical simulations

To simulate the dynamics of our system, we numerically integrate the extended GPE:

$$i\hbar\frac{\partial\psi(\bar{r},t)}{\partial t} = \left[ -\frac{\hbar^2}{2m}\nabla^2 + V_{h.o.}(\bar{r}) + g|\psi(\bar{r},t)|^2 + \int d\bar{r}'V_{dd}(|\bar{r}-\bar{r}'|)|\psi(\bar{r}',t)|^2 \right.$$
$$\left. + \gamma(\varepsilon_{dd})|\psi(\bar{r},t)|^3 \right]\psi(\bar{r},t)$$

in which $V_{h.o.}(\bar{r}) = \frac{1}{2}m(\omega_x^2 x^2 + \omega_y^2 y^2 + \omega_z^2 z^2)$ is the harmonic external potential, $g = \frac{4\pi\hbar^2 a_s}{m}$ is the contact interaction parameter and $V_{dd}(r) = \frac{C_{dd}}{4\pi}\frac{1-3\cos^2\theta}{r^3}$ is the dipolar interaction, with $\theta$ the angle between $r$ and $\hat{z}$ and $C_{dd} = 3\varepsilon_{dd}g$. The last term is the Lee–Huang–Yang energy of quantum fluctuations[53]. Josephson dynamics was induced either by an antisymmetric phase imbalance imprinted with a sinusoidal potential as in the experiment or by an initial antisymmetric population imbalance. Both methods excite the same Josephson normal mode. Atom number and phase for each cluster are calculated at each time step by determining the position of the density minima between the clusters, eliminating their slow and weak oscillations.

The superfluid fraction in Fig. 4a (green dots) is obtained by calculating the coupling energy $K$ in the limit of small initial imbalance ($Z(0) \approx 0.01$), finding values in the range $K \approx k_B(0.1-0.01)$ nK. From equation (5b), we find $N_{34}U \approx k_B(5-7)$ nK, slowly varying with $\varepsilon_{dd}$. The ratio $N_{34}U/(2K)$ is always larger than 25.

## Six-mode Josephson model

We use a set of six-mode Josephson equations with interaction parameters $U_j$, with $j = 1,...,6$ labelling the clusters, five coupling parameters between adjacent clusters $K_{j,j+1}$ and energy offsets $E_0$ and $E_1$ for the opposite side clusters 1 and 6 and 2 and 5, owing to the harmonic trap. We indicate as $K = K_{34}$ and $U = U_3 = U_4$ the coupling and interaction energies, respectively, in two central clusters. The symmetry of the system further allows us to equalize the two side couplings $K' = K_{23} = K_{45}$ and $K'' = K_{12} = K_{56}$ and the two side interactions $U' = U_2 = U_5$ and $U'' = U_1 = U_6$ (see Fig. 2a). We thus have a system of six equations for the time evolution of the populations $N_j$ and five phase differences $\varphi_{ij} = \varphi_i - \varphi_j$:

$$\dot{N}_1 = -2K''\sqrt{N_2 N_1}\sin(\varphi_{21})$$
$$\dot{N}_2 = 2K''\sqrt{N_2 N_1}\sin(\varphi_{21}) - 2K'\sqrt{N_3 N_2}\sin(\varphi_{32})$$
$$\dot{N}_3 = 2K'\sqrt{N_3 N_2}\sin(\varphi_{32}) - 2K\sqrt{N_4 N_3}\sin(\varphi_{43})$$
$$\dot{N}_4 = 2K\sqrt{N_4 N_3}\sin(\varphi_{43}) - 2K'\sqrt{N_5 N_4}\sin(\varphi_{54})$$
$$\dot{N}_5 = 2K'\sqrt{N_5 N_4}\sin(\varphi_{54}) - 2K''\sqrt{N_6 N_5}\sin(\varphi_{65})$$
$$\dot{N}_6 = 2K''\sqrt{N_6 N_5}\sin(\varphi_{65})$$

(5a)

$$\dot{\varphi}_{21} = E_1 + U''N_1 - U'N_2$$
$$\dot{\varphi}_{32} = E_0 + U'N_2 - UN_3$$
$$\dot{\varphi}_{43} = U(N_3 - N_4) \tag{5b}$$
$$\dot{\varphi}_{54} = -E_0 + UN_4 - U'N_5$$
$$\dot{\varphi}_{65} = -E_1 + U'N_5 - U''N_6$$

in which we have considered the case $(N_4 + N_3)U/(2K) \gg 1$ so that we have neglected the tunnelling terms in the evolution of the phases.

In the following, we further consider small-amplitude oscillations such that we can replace $\sqrt{N_iN_j} \approx \sqrt{N_i(0)N_j(0)}$ and $\sin(\varphi_{ij}) \approx \varphi_{ij}$ in equation (5a), in which $N_j(0)$ is the initial population of the $j$th cluster at time $t = 0$. For symmetry reasons, we have $N_1(0) \approx N_6(0)$, $N_2(0) \approx N_5(0)$ and $N_3(0) \approx N_4(0)$. Even in the linear regime, the time evolution of populations and phases predicted by equations (5a) and (5b) shows several frequencies. Harmonic single-frequency oscillations with a $\pi/2$ phase shift between populations and relative phases are observed under the two conditions:

$$\frac{U''}{U'} = 1 + \frac{K'}{K''}\sqrt{\frac{N_3(0)}{N_1(0)}},$$
$$\frac{U'}{U} = \frac{1 + \frac{K}{K'}\sqrt{\frac{N_3(0)}{N_2(0)}}}{1 + \frac{K''}{K'}\sqrt{\frac{N_1(0)}{N_3(0)}}}. \tag{6}$$

In particular, under these conditions, we have

$$\dot{N}_3 - \dot{N}_4 = \alpha(\dot{N}_6 - \dot{N}_1 + \dot{N}_5 - \dot{N}_2), \tag{7}$$

in which $\alpha = 1/(U/U' - U/U'')$. The corresponding Josephson oscillation frequency is

$$\omega_J^2 = 2KU[N_3(0) + N_4(0)]\alpha/(\alpha - 1). \tag{8}$$

To evaluate the parameters in the above equations and verify equation (6), we insert into equations (5a) and (5b) the numerical results for $N_j(t)$ and $\varphi_{ij}(t)$ obtained from GPE simulations. A comparison between GPEs oscillations and the six-mode model is shown in Extended Data Fig. 6. First, the GPE ground state gives $N_3(0) = N_4(0) \approx N/4$, whereas the population of the lateral clusters depends on $\varepsilon_{dd}$. In particular, outer clusters $N_1(0) = N_6(0)$ decrease, whereas $N_2(0) = N_5(0)$ increase as $\varepsilon_{dd}$ increases. The parameters $U$ and $K$ of the central clusters are extracted from equations (4a) and (4b). The other parameters $U'$, $U''$, $K'$ and $K''$ are extracted from fits using equations (5a) and (5b). Overall, we obtain that the interactions parameters are $U/U' \approx 1$, $U/U'' \approx 1/2$ within fluctuations of about 10% for different values of $\varepsilon_{dd}$. On the other hand, the coupling ratio $K/K' \approx 0.6$ is constant, whereas $K/K'' \approx 0.7$ on the BEC side and decreases with $\varepsilon_{dd}$, as do the initial external populations $N_1(0) = N_6(0)$. We thus find that equation (7) is fulfilled and $\alpha = 2$. For this value of $\alpha$, equation (8) gives $\omega_J^2 = 4KU[N_3(0) + N_4(0)]$, in agreement with the main text.

Taking into account equation (7) and the symmetry condition $N_3(0) = N_4(0)$, we find $N_3 - N_4 = \alpha(N_6 - N_1 + N_5 - N_2)$ at each time. We thus have $Z = (\alpha - 1)/\alpha\Delta N/N$, with $\Delta N = N_3 - N_4$. This reduces to $\Delta N = 2NZ$ for $\alpha = 2$. Using equation (5a), we have $\dot{Z} = -4K\sqrt{N_4(0)N_3(0)}/N\sin(\Delta\varphi)$, with $\Delta\varphi = \varphi_{43}$. We can write $2\sqrt{N_3(0)N_4(0)} = N_3(0) + N_4(0) = N_{34}$ and get $\dot{Z} = -2KN_{34}/N\sin(\Delta\varphi)$. The evolution of the phase difference $\Delta\varphi = U(N_3 - N_4)$ (see equation (5b)) rewrites as $\dot{\Delta\varphi} = U\Delta N = 2NUZ$.

It is interesting to take the limit of an infinite array of equal junctions, each one characterized by the same parameters of our central cell. The parts of equation (5a) are all equivalent and, because of the symmetry of the array, $N_i(0) = N_j(0)$ and $\varphi_{i+1,i} = -\varphi_{i,i-1} \, \forall \, i, j$. We then get $\dot{\Delta N} = \dot{N}_i - \dot{N}_{i+1} = -4KN_{i,i+1}\sin(\varphi_{i,i+1})$, with $N_{i,i+1} = N_i(0) + N_{i+1}(0)$, equivalent to equation (4a). Equation (4b) for the phase evolution applies in the infinite case as well.

## Data availability

All data of the figures in the manuscript and Methods are available in a Zenodo repository at https://doi.org/10.5281/zenodo.10045059 (ref. 54). Source data are provided with this paper.

## Code availability

The codes that support the findings of this paper are available from the corresponding authors on reasonable request.

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

**Acknowledgements** Financed by the European Union (ERC, SUPERSOLIDS, no. 101055319) and by the QuantERA Programme that has received funding from the European Union's Horizon 2020 research and innovation programme under grant agreement nos. 731473 and 101017733, projects MENTA, SQUEIS and MAQS, with funding organization Consiglio Nazionale delle Ricerche. We acknowledge financial support from PNRR MUR project PE0000023-NQSTI financed by the European Union - Next Generation EU. We gratefully acknowledge technical assistance from A. Barbini, A. Hajeb, F. Pardini, M. Tagliaferri and M. Voliani.

**Author contributions** N.A., G.B., A.F., C.G., L.T. and G.M. performed the experimental measurements. B.D., L.P. and A.S. developed the theoretical model and carried out the numerical simulations. N.A., G.B., B.D. and L.P. performed the experiment–theory comparison. N.A., G.B., B.D., L.P., A.S., A.F., M.F., C.G., M.I., L.T. and G.M. contributed to the interpretation of the results and the writing of the paper.

**Competing interests** The authors declare no competing interests.

**Additional information**
**Correspondence and requests for materials** should be addressed to A. Smerzi or G. Modugno.

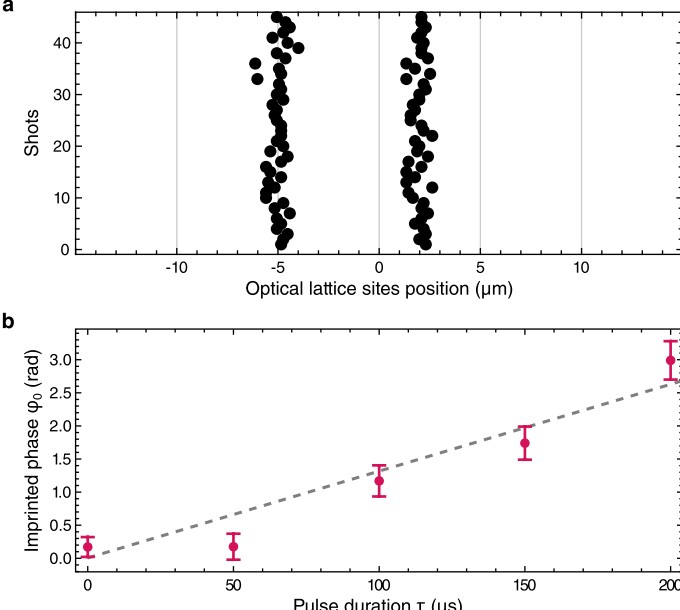

**Extended Data Fig. 1 | Characterization of the optical lattice. a**, Stability of the lattice. Black dots are the relative positions of the density peaks of a BEC loaded into the optical lattice with respect to the average centre of mass, for 45 different measurements. The s.d. of fluctuations for each lattice position is $\sigma_{lattice} \approx 0.35\,\mu m$. **b**, Calibration of the initial phase difference imprinted on the two main clusters as a function of the lattice pulse duration. Red dots are experimental data obtained by imprinting the optical lattice potential for different pulse durations. Error bars are the s.e.m. of 15–20 data points. The dashed line is the theoretical prediction ($U\tau/\hbar$) with a lattice depth $U = 100$ nK.

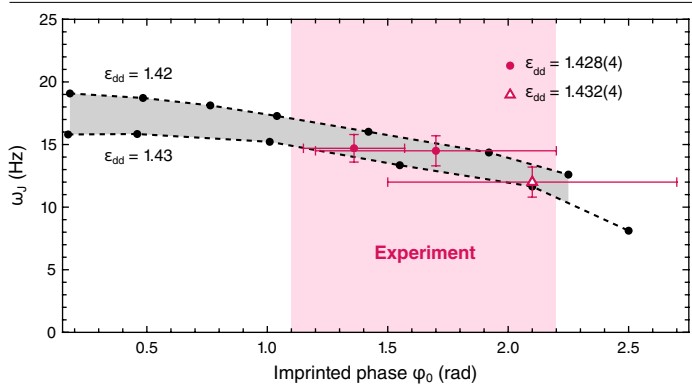

**Extended Data Fig. 2 | Josephson frequency as a function of the phase amplitude.** Black dots are theory, red dots are experiment. Vertical error bars are extracted from the sinusoidal fit of the Josephson oscillations. Horizontal error bars are the s.e.m. of the phase difference detected at $t = 0$, after the phase imprinting. The highlighted region is the relevant one for the experiment. For typical experimental amplitudes $\pi/2$, we observe a frequency reduction of about 15% compared with the small-excitation regime.

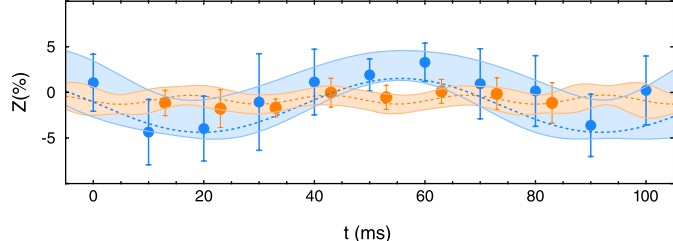

**Extended Data Fig. 3 | Absence of Josephson oscillations for a standard BEC.**
Time evolution of the population imbalance after the phase imprinting for a
supersolid (blue) and an unmodulated BEC (orange), extracted using optical
separation. Dashed lines are sinusoidal fits with one s.d. confidence bands in
shaded colour. The fitted oscillation amplitude of the BEC is compatible with
zero. Error bars are the s.e.m. of 15–20 data points.

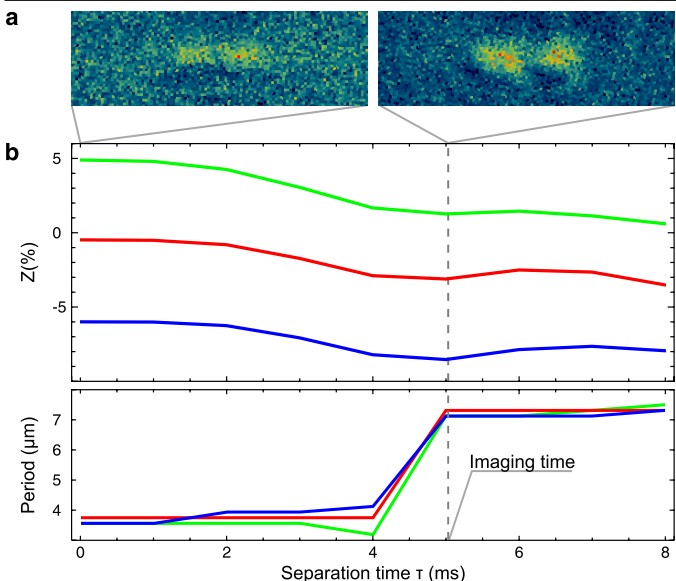

**b**

Z(%)

Period (μm)

Imaging time

Separation time τ (ms)

**Extended Data Fig. 4 | Analysis of the optical-separation technique.**
**a**, Experimental in situ images of two balanced ($Z = 0$) supersolids without any manipulation (left) and using optical separation (right). **b**, Numerical simulation of the dynamics of three different supersolids with initial population imbalance $Z_0 = 5\%$ (green), $Z_0 = 0\%$ (red) and $Z_0 = -5\%$ (blue), during optical separation. Top row, imbalance $Z$. Bottom row, distance between the central clusters.

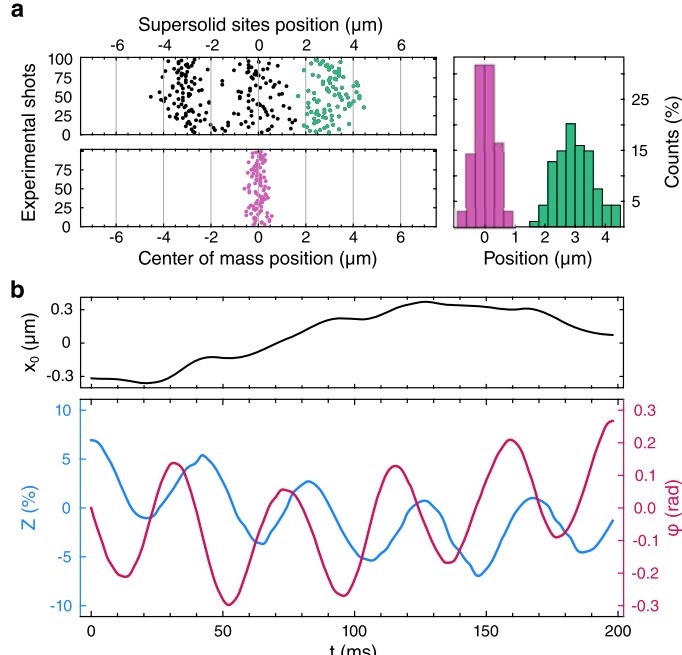

**Extended Data Fig. 5 | Evidence of the Goldstone mode in experimental and numerical data. a**, Left panels, fluctuation of the clusters positions (black and green points) and centre of mass (pink points) of the supersolid for about 100 experimental shots. Right panel, histograms of the right-cluster (green) and of the centre-of-mass (pink) positions. **b**, Simulation of the Josephson dynamics coupled to the Goldstone oscillation. The position of the density minimum $x_0$ (top row) shows a clear oscillation at the Goldstone frequency $\omega_G \ll \omega_J$. This lower frequency also appears in $Z$ and $\Delta\varphi$ (bottom row) on top of the standard Josephson dynamics.

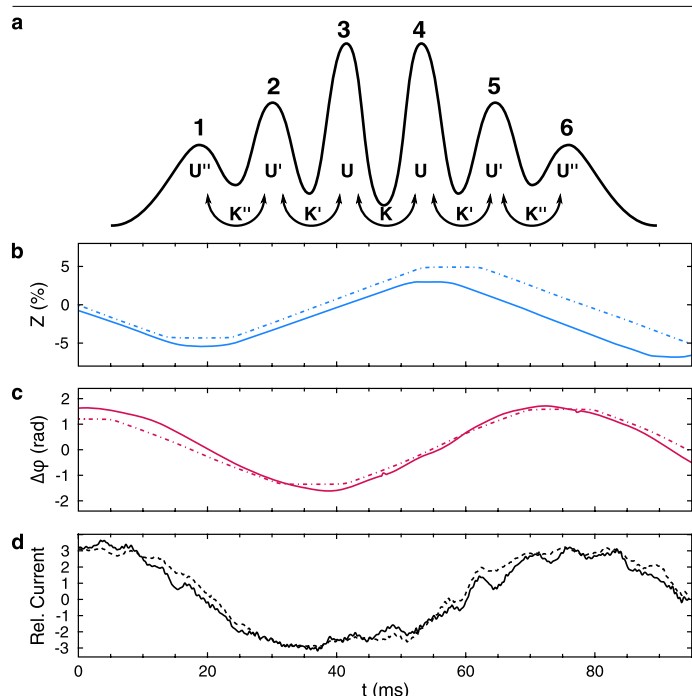

**Extended Data Fig. 6 | Six-mode model and numerical simulations. a**, Sketch of the inhomogeneous system with six clusters with interaction energies $U$, $U'$ and $U''$, with coupling energies $K$, $K'$ and $K''$. The modes in the sketch are not to scale (compare with the simulation in Fig. 2a). **b,c**, Comparison between the time evolution calculated from the GPE simulations (solid lines) and from the six-mode model (dot-dashed lines) for $Z$ (**b**) and $\Delta\varphi$ (**c**). **d**, Relative currents between the central and lateral clusters appearing in equation (7), from GPE simulations. The solid line is $(\dot{N}_3 - \dot{N}_4)/2$ and the dashed line is $(\dot{N}_6 + \dot{N}_5 - \dot{N}_2 - \dot{N}_1)$.