## [Peer Review File · Nature]

Manuscript Title: Measurement of the superfluid fraction of a supersolid by Josephson effect

Reviewer Comments & Author Rebuttals

Reviewer Reports on the Initial Version:

Referees' comments:

Referee #1 (Remarks to the Author):

The authors study the excitation of a trapped elongated dipolar supersolid by imprinting a periodic phase modulation using an off-resonant standing wave potential. The period of the standing wave is chosen to be twice the spatial period of the supersolid. Josephson-like oscillations in the phase and population imbalance are measured using real space and time of flight imaging. In addition, numerical calculations using extended Gross-Pitaevskii equation are carried out.

The Josephson frequency is measured as a function of the relative interaction parameter and the coupling energy is determined with additional input from numerical simulations. The ratio of this coupling energy and the kinetic energy of a single cell is plotted and considered as the superfluid fraction of the supersolid.

The measurement of collective excitations in a supersolid is an ongoing and important topic. In particular, the soft Roton modes are considered as a characteristic feature of a supersolid [see recent work of the groups looking at dipolar supersolids]. In a related work, the response of a dipolar supersolid to an interaction quench has been analysed in terms of a Josephson junction array model [doi.org/10.1038/s41567-020-01100-3]. In the present work, the excitation is carried out using the phase twisting caused by the optical lattice potential. Another novel aspect of the present work is the interpretation of the Josephson oscillations in terms of a superfluid fraction. Indeed, it is argued in the first paragraph that the superfluid fraction, which measures the reduction in superfluid stiffness due to the spatial modulation, is a key property of the supersolid.

The concept of superfluid stiffness is well introduced in the beginning of the paper, referencing to Leggett's work. Inspired by Leggett's work an alternative expression for the superfluid fraction is proposed, which is the ratio of the Josephson coupling energy and the kinetic energy across a single cell. The key question of the paper is, whether this proposal is valid. My concern is that considering only a single cell is not a convincing criterium for the specific superfluid properties of a supersolid, whose very characteristic is the long-range density periodicity. Just looking at the response to a phase twist across a single cell is not persuasive.

In general, the paper is well written. However, for many quantities the errors are neither stated in the main text nor the supplement.

In conclusion, I do not recommend publication in Nature. The main reason for this is the short spatial range of the observable for the superfluid fraction, whereas the supersolid is characterized by its long-ranged order. This discrepancy is not resolved for the reader of the present manuscript.

Referee #2 (Remarks to the Author):

The manuscript describes a method to measure the superfluid fraction of a supersolid using the Josephson effect. The system is a cold atomic dipolar gas (bosonic dysprosium) in an anisotropic harmonic trap. Similar system had been studied previously by some of the authors in Ref. 7, but the motivation and questions raised in the current manuscript are fundamentally different. The system displays “supersolidity,” i.e. non-uniform density in the elongated direction.

An essential finding is the experimental identification of a single frequency response associated with variations of density and phases upon imprinting a phase difference between nearby clusters. This frequency is identified with a Josephson response and fitted with theoretical predictions from a GP equation and a six-mode Josephson model, which guided the determination of the superfluid fraction. The methods to model the system and the results are sound. This work represents a significant experimental advance and merits publication.

In solid-state systems, non-uniform condensates have received significant attention, such as pair-density waves observed in several classes of complex materials. This work may shed new light on pdw states, though the degrees of freedom and experimental conditions are quite distinct, one example being the presence of a symmetry-breaking external harmonic trap and what effects it entails. Nevertheless, this work opens an interesting door to explore non-uniform unconventional condensates.

Referee #3 (Remarks to the Author):

The paper on “Sub-unity superfluid fraction of a supersolid from self-induced Josephson effect” reports new and highly intriguing data on the new and hot research topic of dipolar supersolids, that has recently attracted significant attention in the ultra-cold atom research community. Albeit clear signals of the long-sought supersolid state of matter were already reported experimentally and the existence of a supersolid state of matter was for the first time clearly established in dipolar quantum gases (refs 7-9 in this article), many important questions were left open. Among them is the quest for a clear smoking-gun signal of the supersolidity, which is addressed in this paper via the Josephson effect, in a novel and interesting manner. What makes this work particularly relevant for this field is, that it offers new ways to quantify the degree of supersolidity in these novel systems, opening the path to new studies of dynamical properties of such novel ultra-cold quantum matter. It is shown that a supersolid can sustain phase-density oscillations, in similarity to Josephson junction arrays. The data shown and conclusions drawn allow a new approach to determine the superfluid

fraction in novel dipolar supersolids, which will be important for future applications of such systems. Because of this, in my opinion the work deserves to be published in Nature. However, the self-induced nature of the Josephson junction physics in such systems is not new and has been discussed previously in the literature; most importantly, in Ref. 29 of the present manuscript, which is cited only very late in the paper, and (irritatingly enough) it is not stated clearly when credit is claimed for the novelty of the idea. This must be changed, and appropriate credit be given to the work of Ilzhofer et al. (see also my comments further below). At the end of the abstract it says "Our work also discloses a new type of Josephson junction". However, the basic idea has been around before; see the mentioned reference and related works cited in ref. 29.

The data shown and simulations appear sound, and I could not spot any inconsistencies or mistakes, neither in experimental data nor theoretical modelling (which is standard extended Gross-Pitaevskii). The presentation is ok but at various places in the paper some amendments may help to make the paper easier to read; I make some more definite suggestions below.

Overall, the conclusions appear valid and robust, and I have no reason to doubt the validity of the work.

Regarding references and appropriate credit to previous work, most importantly I believe that better credit needs to be given to the work of Ilzhofer et al., as discussed above. In the present version of the manuscript this work appears under Ref. 29. Yet, the novelty of the self-induced Josephson junctions is claimed, in disagreement with the reference 29 which is only cited at a later stage in this manuscript. It must be clearly mentioned right in the beginning when the novelty of discussing self-induced Josephson junctions is claimed.

Regarding clarity and context, including abstract and summary, I have a few remarks. Right in the beginning of the paper a lump of references 1-12 is thrown on the reader; this is useless without proper context. Likewise, the sentence "They might all be related..." does not help the reader to get interested in the work. I suggest avoiding the pile-up of references but introduce what a supersolid is, and why it is interesting and what is novel about it. A short history of the experimental discovery of supersolidity in dipolar quantum gases is necessary at this point, but it is not given in the present version of the text. Please clean-up and give more accurate credit to the works leading to the experimental discovery of supersolidity in the dipolar case. On page 2, a large number of possibly related quantum phases and systems is listed, and it is stated that "all these systems might be related to supersolidity". This vague statement should be removed; alternatively, the examples given must be put into a better context.

On page 3, it is claimed that a "paradigm shift" is made in this work, trying to boost the paper. Please remove this unnecessary and overdriven statement.

Finally, I am wondering why a harmonic, elliptic confinement was chosen for this study, which brings along the additional complication that the supersolid droplet crystal is asymmetric, concerning the norm of the next-to center droplets and those close to the boundary. A better choice would be a box potential or, ideally, a ring confinement? All droplets would then be identical with $Z=1$.

A comparison is made between a Josephson self-induced junction lattice and a homogeneous superfluid, and it is concluded that the observed oscillations are clearly related to the formation of the intrinsic junction lattice. I wonder, has this been compared to the action of a periodic optical lattice in the background? How different are the oscillations in the latter case from the self-induced Josephson junction array?

Some minor suggestions:

- 1) In the introduction you say that the superfluid fraction has not been assessed in the supersolid case. This is not generally true, as there are some theory works that precisely address this point. What is probably meant here is that this has not yet been assessed experimentally. Please change the text accordingly and give the proper references.
- 2) On page 2, where it says "As shown in Fig. 1" it would be better to say "as sketched in Fig. 1" as the plot is schematic rather than containing any data.
- 3) On page 3 the "individual cells of a supersolid" are discussed but it is not clear how these cells are defined? Please be more precise.
- 4) On page 3, when the self-induced Josephson oscillation are discussed, please include again Ref. 29.
- 5) On page 5, when referring to a dipolar supersolid only reference 7 is cited. Please cite also 8 and 9, as you did at other places, for the discovery of supersolidity in dipolar systems.
- 6) On page 5 you state that "interactions are weak, allowing accurate theoretical modeling". This is only partially true, the applied extended Gross-Pitaevskii method is still only approximate. This should be mentioned.
- 7) On page 6 you speak about the "asymmetry between the left and right halves". Please be more precise; asymmetry of what quantity? Norm?
- 8) I understood that U is proportional to the interaction energy, but where is it defined in the text? I got confused about this point when I came to Eq. 4b
- 9) Also, the discussion of the quantity Z and its definition should come earlier in the text.
- 10) On page 7 you mention "transverse effects" but to the reader it is not obvious what these are; please explain.

Referee #1 Response

The authors study the excitation of a trapped elongated dipolar supersolid by imprinting a periodic phase modulation using an off-resonant standing wave potential. The period of the standing wave is chosen to be twice the spatial period of the supersolid. Josephson-like oscillations in the phase and population imbalance are measured using real space and time of flight imaging. In addition, numerical calculations using extended Gross-Pitaevskii equation are carried out.

The Josephson frequency is measured as a function of the relative interaction parameter and the coupling energy is determined with additional input from numerical simulations. The ratio of this coupling energy and the kinetic energy of a single cell is plotted and considered as the superfluid fraction of the supersolid.

The measurement of collective excitations in a supersolid is an ongoing and important topic. In particular, the soft Roton modes are considered as a characteristic feature of a supersolid [see recent work of the groups looking at dipolar supersolids]. In a related work, the response of a dipolar supersolid to an interaction quench has been analysed in terms of a Josephson junction array model [doi.org/10.1038/s41567-020-01100-3]. In the present work, the excitation is carried out using the phase twisting caused by the optical lattice potential. Another novel aspect of the present work is the interpretation of the Josephson oscillations in terms of a superfluid fraction. Indeed, it is argued in the first paragraph that the superfluid fraction, which measures the reduction in superfluid stiffness due to the spatial modulation, is a key property of the supersolid.

We thank Referee #1 for their careful reading and appreciation of our work, and for the constructive comments.

In general, the paper is well written. However, for many quantities the errors are neither stated in the main text nor the supplement.

We thank the Referee for this note. In the revised manuscript, all relevant figure captions explain that the error bars represent statistical uncertainties, either resulting from multiple measurements or from the error propagation in the successive analysis.

The concept of superfluid stiffness is well introduced in the beginning of the paper, referencing to Leggett's work. Inspired by Leggett's work an alternative expression for the superfluid fraction is proposed, which is the ratio of the Josephson coupling energy and the kinetic energy across a single cell. The key question of the paper is, whether this proposal is valid.

My concern is that considering only a single cell is not a convincing criterium for the specific superfluid properties of a supersolid, whose very characteristic is the long-range density periodicity. Just looking at the response to a phase twist across a single cell is not persuasive.

In conclusion, I do not recommend publication in Nature. The main reason for this is the short spatial range of the observable for the superfluid fraction, whereas the supersolid is characterized by its long-ranged order. This discrepancy is not resolved for the reader of the present manuscript.

We thank the Referee for raising this important point on which we realise that we were not sufficiently clear. The question is whether our analysis and measurements based on individual cells of a supersolid, which has a long-range density periodicity, can provide the correct superfluid fraction. The short answer is the following. Since the wavefunction of an ideal supersolid is periodic, if also the dynamics that one studies has the same periodicity, it is formally correct to extract the superfluid fraction from the phase twist on a single cell instead of the global phase twist, as originally found by Leggett. Let us elaborate more on this point, because there are important differences between an ideal supersolid and the inhomogeneous supersolid we have in the experiment.

The superfluid fraction is a global property, but due to the very same long-range density periodicity of an ideal supersolid with identical lattice cells, one can define it on a single cell provided that the dynamics under study has the same periodicity of the lattice. The superfluid fraction defined on a single cell therefore coincides with the global one. The first author to recognize this was A. J. Leggett in 1970, who considered the rotation sketched in Fig. 1b. In ref. 16 he calculated on a single cell eq. 2 as the upper bound for the superfluid fraction of a ring-shaped system. Almost all successive works have followed the same strategy, defining the superfluid fraction on a single cell. See for example these more recent studies of the superfluid fraction and related properties in 1D and 2D supersolids:

Nonclassical rotational inertia fraction in a one-dimensional model of a supersolid, N. Sepúlveda, C. Josserand, and S. Rica, Phys. Rev. B 77, 054513 (2008).

Superfluid density in a two-dimensional model of supersolid, N. Sepúlveda, C. Josserand, and S. Rica, Eur. Phys. J. B 78, 439–447 (2010).

Supersolidity around a critical point in dipolar Bose-Einstein condensates, Y.-C. Zhang, F. Maucher, and T. Pohl, Phys. Rev. Lett. 123, 015301 (2019).

Supersolidity and crystallization of a dipolar Bose gas in an infinite tube, J. C. Smith, D. Baillie, and P. B. Blakie, Phys. Rev. A 107, 033301 (2023).

In our work we follow the same approach, and with eq. 1 and eq. 3 we define on a single lattice cell the superfluid fraction of an ideal, truly periodic supersolid as in Fig. 1. As sketched in Fig.1c, in the Josephson oscillation mode we introduce with this work the phase twist changes sign from cell to cell, differently from the global rotation of Fig. 1b. However, the sign of the phase gradient is irrelevant since the kinetic energy contains the square of the phase gradient. Therefore, also for Josephson oscillations it is possible to define the superfluid fraction on a single cell.

The supersolid we study experimentally is not homogeneous as in the theory, see Fig. 2 and the related discussion. The density modulation is larger at the center and decreases going towards the edges. Therefore, although the excitation we provide to the system with the optical lattice is periodic, the tunneling energy of the central cell measured or calculated through eq. 3 does not provide the overall superfluid fraction of our inhomogeneous system. We interestingly found that it provides instead the superfluid fraction of an ideal supersolid composed by lattice cells identical to our central one. We derived this key conclusion with the six-mode model presented in the manuscript, which takes into account the global dynamics in our inhomogeneous system. We discussed the results of the model in the central part of the original manuscript:

“... there exists a normal mode of the system where the dynamical variables of the two central clusters of the supersolid decouple from the lateral ones. This results in Josephson-like oscillations described by the two coupled equations ... 4(a-b) that are equivalent to those of a simple pendulum with angle $\Delta\varphi$ and angular momentum ΔN , and in the small-angle limit feature sinusoidal oscillations with a single frequency, $\omega_j = \sqrt{4KUN_{34}}$. We emphasise that the current-phase relation Eq.(4a) as well as ω_j^2 differ by a factor 2 with respect to the Josephson equations of two weakly coupled Bose-Einstein condensates, due to the contribution of the lateral clusters. Notice also that Eqs. 4(a-b) depend only on the coupling energy K and the interaction energy U of the two central clusters, in contrast to the expectation that the inhomogeneity of the trapped system may introduce other energies in the equations of motion. We checked by Gross-Pitaevskii simulations that our experimental configuration satisfies the conditions to have a Josephson-like normal mode ...”

This analysis is relevant in the context of the point raised by the Referee, because it provides a connection between the dynamics of the central cell leading to the superfluid fraction, and the long-range density modulation of the supersolid. We indeed derived for our central cell a current-phase relation (Eq. 4a) formally identical to the case of a lattice cell of an infinite and homogeneous density-modulated system. Since we find that the central cell

behaves as if the supersolid was homogeneous, our results for the superfluid fraction are very general. We note that the superfluid fraction of the inhomogeneous system in the experiment, which we did not attempt to measure, would not have the same general interest.

In the original manuscript, we discussed the point raised by the Referee in the short paragraph just above Fig. 4: “What we measure here is the superfluid fraction of the central cell of our inhomogeneous supersolid, for a one-dimensional phase twist. Such a local quantity would coincide with the global superfluid fraction of a hypothetical homogeneous supersolid, composed of cells identical to our central one”.

The comment by the Referee was helpful to understand that such a key concept should be explained earlier and better. Therefore, in the revised manuscript we have removed the sentence above Fig. 4 and added two separate sentences.

The first new sentence is just after eq. 2: “Note that the reduction of the superfluid fraction, which is a global property, to a single lattice cell is possible due to the periodicity of the wavefunction of the supersolid ¹⁶.”

The second new sentence is just above Fig. 2: “Since our system is inhomogeneous, we focus our attention on the central cell, the one delimited by the clusters 3 and 4 in Fig. 2. As we will show, the superfluid fraction we derive from that cell corresponds to the superfluid fraction of a hypothetical homogeneous supersolid with all cells identical to the central one, as in Fig. 1, which is the system of general interest.”

Moreover, we added a note in the explanation following Eqs. 4a-b: “We emphasise that the current-phase relation Eq.(4a) as well as ω_j^2 differ by a factor 2 with respect to the Josephson equations of two weakly coupled Bose-Einstein condensates, due to the contribution of the lateral clusters, *but are equal to the ones of a hypothetical homogeneous supersolid* “. This point is explained more in detail by a new paragraph at the end of the Methods.

We also made the discussion in the Methods about the six-mode model clearer, by adding at the end the following paragraph: “It is interesting to take the limit of an infinite array of equal junctions, each one characterised by the same parameters of our central cell. Eqs. 5(a) are all equivalent, and due to the symmetry of the array, $N_i(0) = N_j(0)$ and $\varphi_{i+1,i} = -\varphi_{i,i-1} \forall i, j$.

We then get $\Delta\dot{N} = \dot{N}_i - \dot{N}_{i+1} = -4KN_{i,i+1} \sin(\varphi_{i,i+1})$, with $N_{i,i+1} = N_i(0) + N_{i+1}(0)$, equivalent to Eq. (4a) in the main text. Eq. (4b) for the phase evolution applies in the infinite case as well.”

Finally, we removed the term “local” from the statement in the summary: “... we show that from the current-phase dynamics we can derive directly the local superfluid fraction”, and the term “of the supersolid” from the title of the caption of Fig. 4: “Superfluid fraction of the supersolid from Josephson oscillations”. We now realise that both terms were not appropriate.

We hope that with these modifications it will be now clear to the readers why a local measurement of Josephson oscillations can provide the global superfluid fraction.

Referee #2 Response

The manuscript describes a method to measure the superfluid fraction of a supersolid using the Josephson effect. The system is a cold atomic dipolar gas (bosonic dysprosium) in an anisotropic harmonic trap. Similar system had been studied previously by some of the authors in Ref. 7, but the motivation and questions raised in the current manuscript are fundamentally different. The system displays “supersolidity,” i.e. non-uniform density in the elongated direction.

An essential finding is the experimental identification of a single frequency response associated with variations of density and phases upon imprinting a phase difference between nearby clusters. This frequency is identified with a Josephson response and fitted with theoretical predictions from a GP equation and a six-mode Josephson model, which guided the determination of the superfluid fraction. The methods to model the system and the results are sound. This work represents a significant experimental advance and merits publication.

In solid-state systems, non-uniform condensates have received significant attention, such as pair-density waves observed in several classes of complex materials. This work may shed new light on pdw states, though the degrees of freedom and experimental conditions are quite distinct, one example being the presence of a symmetry-breaking external harmonic trap and what effects it entails. Nevertheless, this work opens an interesting door to explore non-uniform unconventional condensates.

We thank Referee #2 for their careful reading and appreciation of our work, and for the constructive comments.

Regarding the comment by the Referee on “the presence of a symmetry-breaking external harmonic trap”, with our analysis based on the six-mode model we could directly connect the results for our inhomogeneous supersolid in the harmonic trap to the ideal case of a supersolid with homogeneous density modulation. Please see the response to Referee #1. Therefore, we think that our results are directly applicable to the other condensed-matter systems that are not harmonically confined.

Referee #3 Response

The paper on “Sub-unity superfluid fraction of a supersolid from self-induced Josephson effect” reports new and highly intriguing data on the new and hot research topic of dipolar supersolids, that has recently attracted significant attention in the ultra-cold atom research community. Albeit clear signals of the long-sought supersolid state of matter were already reported experimentally and the existence of a supersolid state of matter was for the first time clearly established in dipolar quantum gases (refs 7-9 in this article), many important questions were left open. Among them is the quest for a clear smoking-gun signal of the supersolidity, which is addressed in this paper via the Josephson effect, in a novel and interesting manner. What makes this work particularly relevant for this field is that it offers new ways to quantify the degree of supersolidity in these novel systems, opening the path to new studies of dynamical properties of such novel ultra-cold quantum matter. It is shown that a supersolid can sustain phase-density oscillations, in similarity to Josephson junction arrays. The data shown and conclusions drawn allow a new approach to determine the superfluid fraction in novel dipolar supersolids, which will be important for future applications of such systems. Because of this, in my opinion the work deserves to be published in Nature. However, the self-induced nature of the Josephson junction physics in such systems is not new and has been discussed previously in the literature; most importantly, in Ref. 29 of the present manuscript, which is cited only very late in the paper, and (irritatingly enough) it is not stated clearly when credit is claimed for the novelty of the idea. This must be changed, and appropriate credit be given to the work of Ilzhofer et al. (see also my comments further below). At the end of the abstract it says “Our work also discloses a new type of Josephson junction”. However, the basic idea has been around before; see the mentioned reference and related works cited in ref. 29.

The data shown and simulations appear sound, and I could not spot any inconsistencies or mistakes, neither in experimental data nor theoretical modelling (which is standard extended Gross-Pitaevskii). The presentation is ok but at various places in the paper some amendments may help to make the paper easier to read; I make some more definite suggestions below.

Overall, the conclusions appear valid and robust, and I have no reason to doubt the validity of the work.

We thank Referee #3 for their careful reading and appreciation of our work, and for the constructive comments. Please see below our response to the remark on ref. 29.

Regarding references and appropriate credit to previous work, most importantly I believe that better credit needs to be given to the work of Ilzhofer et al., as discussed above. In the present version of the manuscript this work appears under Ref. 29. Yet, the novelty of the self-induced Josephson junctions is claimed, in disagreement with the reference 29 which is only cited at a later stage in this manuscript. It must be clearly mentioned right in the beginning when the novelty of discussing self-induced Josephson junctions is claimed.

We thank the Referee for raising this point. The work reported in ref. 29 is certainly noteworthy in the context of our work, as it introduced a Josephson model for the equilibration dynamics of a supersolid, as noted also by Referee #1. However, we would like

to remark that the work in ref. 29 did not provide any experimental or theoretical evidence of the Josephson current-phase dynamics in supersolids as we do in our work. Since this is an important point, we will discuss it in some detail.

For sake of clarity, let us report here below the key statements of ref. 29, starting from title and abstract:

“Phase coherence in out-of-equilibrium supersolid states of ultracold dipolar atoms, by P. Ilzhof et al., Nature Physics 2021.

A supersolid is a counterintuitive phase of matter that combines the global phase coherence of a superfluid with a crystal-like self-modulation in space. Recently, such states have been experimentally realized using dipolar quantum gases. Here we investigate the response of a dipolar supersolid to an interaction quench that shatters the global phase coherence. We identify a parameter regime in which this out-of-equilibrium state rephases, indicating superfluid flow across the sample as well as an efficient dissipation mechanism. We find a crossover to a regime where the tendency to rephase gradually decreases until the system relaxes into an incoherent droplet array. Although a dipolar supersolid is, by its nature, ‘soft’, we capture the essential behaviour of the de- and rephasing process within a rigid Josephson junction array model. Yet, both experiment and simulation indicate that the interaction quench causes substantial collective mode excitations that connect to phonons in solids and affect the phase dynamics.”

The introduction of ref. 29 does not mention the Josephson effect. The Josephson model is introduced on page 2: “Our system of multiple superfluid droplets with individual phases, interconnected via weak links, is reminiscent of a Josephson junction array (JJA)³⁴. Motivated by this analogy, we investigate whether a simple JJA model can adequately describe the observed phase dynamics.” One later understands that the relevant parameters of the Josephson Hamiltonian are deduced phenomenologically from the observed dynamics of the rephasing process.

It is important to note that the measurements of ref. 29 are carried out only after a free expansion and only to probe the amount of phase coherence across the whole array. No measurement of a deterministic phase dynamics, traceable to Josephson oscillations, is performed. Moreover, the population imbalance Z , the conjugate variable to the phase difference ϕ in the Josephson effect, is not measured. Any observation of Josephson oscillations must include measurements of both the conjugate variables.

Here are finally the conclusions of ref. 29: “In conclusion, we have performed a study of the out-of-equilibrium dynamics of a dipolar supersolid after an interaction-driven phase excitation that fully destroys its phase coherence. We have demonstrated that if the inter-droplet density links are sufficient, this phase-scrambled system relaxes into an equilibrium phase-coherent state. With decreasing link strength, the rephasing substantially slows down and eventually ceases in the ID regime. We find an overall consistency between the phase dynamics observed in the experiment and an intuitive, theoretically easily tractable rigid JJA model. However, both ab initio simulations and experimental observations indicate post-quench collective excitations of the droplet array, which can affect the phase dynamics. Our study shows the evidence of particle flow across a dipolar supersolid,

connecting to its superfluidity. It also suggests the efficient dissipation of phase excitations, whose microscopic mechanism is still under question. Future experimental works, combined with advanced out-of-equilibrium theoretical models, will be crucial to understand the relaxation dynamics at play in isolated and open supersolid states of quantum matter.”

Therefore, we can conclude that the claim of ref. 29 with respect to the Josephson effect is limited to “the overall consistency between the phase dynamics observed in the experiment and an intuitive, theoretically easily tractable rigid JJA model.” Throughout ref. 29, no claims have been made instead of having discovered the Josephson effect in the supersolid, and concepts like “self-induced Josephson oscillations or self-induced Josephson junctions” are not mentioned.

Having clarified this aspect, we now answer the specific remarks by the Referee.

We agree with the Referee that the modelling of the supersolid in terms of a Josephson junction array deserves more visibility already at the level of the summary paragraph. We therefore modified the 7th paragraph of the summary, which now reads: “A supersolid can be modelled as an array of Josephson junctions²⁰, due to the spontaneous spatial modulation of the wavefunction”. Ref. 29 of the original manuscript becomes therefore ref. 20 in the revised manuscript.

We think instead that the place in the introduction where we discuss the new ref. 20 in slightly more detail is the correct one, i. e. after having introduced the Josephson effect, otherwise the reader won't understand. We have slightly changed the construction of the sentence, to be more clear: “So far, the analogy between a supersolid and an array of Josephson junctions has been only employed to model the relaxation towards the ground state of a dipolar supersolid²⁰.”

To make clearer that our work is the first one to prove both experimentally and theoretically the existence of Josephson oscillations in a supersolid, we have slightly revised the sentence: “Here we demonstrate *experimentally and theoretically* that a supersolid can in fact sustain coherent phase-density oscillations, behaving as an array of Josephson junctions.”

The Referee had a remark also on the final sentence of the summary paragraph: “At the end of the abstract it says “Our work also discloses a new type of Josephson junction”. However, the basic idea has been around before; see the mentioned reference and related works cited in ref. 29”. In our intention, that statement was not referring to the “self-induced Josephson junctions” in general. It was instead pointing to the novel properties of the junctions in a supersolid in terms of their capability of moving through the Goldstone mode, a phenomenon yet to be studied, as we briefly discussed in the conclusions and in the methods. We now realise the possibility of misunderstanding such a sentence, so we prefer to remove it.

In addition, from the sentence in the conclusions "The demonstration of self-sustained Josephson oscillations in a supersolid not only establishes an analogy between supersolids and Josephson junction arrays, but also provides a novel proof of the extraordinary nature of supersolids compared to ordinary superfluids and crystals", we removed the statement "establishes an analogy between supersolids and Josephson junction arrays", which was incorrect since the analogy had been established by ref. 29. The rest of the sentence is correct.

Regarding clarity and context, including abstract and summary, I have a few remarks. Right in the beginning of the paper a lump of references 1-12 is thrown on the reader; this is useless without proper context. Likewise, the sentence "They might all be related..." does not help the reader to get interested in the work. I suggest avoiding the pile-up of references but introduce what a supersolid is, and why it is interesting and what is novel about it. A short history of the experimental discovery of supersolidity in dipolar quantum gases is necessary at this point, but it is not given in the present version of the text. Please clean-up and give more accurate credit to the works leading to the experimental discovery of supersolidity in the dipolar case.

This point by the Referee deals with the summary paragraph at the beginning of the manuscript, where the length constraints do not allow extensive explanations. Our original text reads: "Recently, a novel class of superfluids and superconductors with spontaneous spatially-periodic modulation of the superfluid density has been discovered in a variety of systems¹⁻¹². They might all be related to the concept of a supersolid phase of matter, which has by definition a macroscopic wavefunction with spatial modulation originating from the simultaneous, spontaneous breaking of the gauge and translational symmetries¹³⁻¹⁶." In preparing our manuscript, we made an effort to communicate the generality of our findings to the widest possible audience, including all the condensed-matter physics sectors where supersolid-like phases are under investigation, in different types of superconductors and superfluids. Given the limited space, we cannot introduce each system here (we do it a few sentences below, in the introduction). That's why we decided to describe all the types of system under study (refs. 1-12) as "superfluids and superconductors with spontaneous spatially-periodic modulation of the superfluid density".

Regarding the introduction to supersolids, we think that our definition as the phase that has "a macroscopic wavefunction with spatial modulation originating from the simultaneous, spontaneous breaking of the gauge and translational symmetries", is correct and complete, although it is concise.

Regarding the short history of the experimental discovery of supersolidity in dipolar quantum gases, there is unfortunately no space available even for a minimal discussion, also given the generality of the summary. We therefore implemented the suggestion by the Referee by adding a citation to a recent review of the experimental work on dipolar quantum gases, including the work leading to the discovery of supersolidity: L. Chomaz, I. Ferrier-Barbut, F. Ferlaino, B. Laburthe-Tolra, B. L. Lev and T. Pfau, Dipolar physics: a review of experiments with magnetic quantum gases, Rep. Prog. Phys. 86 026401 (2022). The citation is added after eq. 3, where we discuss for the first time dipolar supersolids in detail: "... in a dipolar

supersolid⁷⁻⁹. This system is particularly appealing to study fundamental aspects of supersolidity³⁸: ...”

On page 2, a large number of possibly related quantum phases and systems is listed, and it is stated that “all these systems might be related to supersolidity”. This vague statement should be removed; alternatively, the examples given must be put into a better context.

We followed this suggestion by revising that sentence into “*The periodic structure of the wavefunction of all these systems is a prerequisite for supersolidity*, which however so far has emerged clearly only in some cold-atom systems with the evidence of the double spontaneous symmetry breaking and of the mixed superfluid-crystalline character”. We think that this explains quite clearly why all the “quantum phases with spontaneous modulation of the wavefunction ... under study in a variety of bosonic and fermionic systems” that we list are related to supersolids. We added one additional reference to a recently discovered magnetic system with supersolid-like properties on the spin degrees of freedom, see ref.22, J. Xang et al, *Nature* 625, 270–275 (2024). This is described in this part of the introduction among the systems related to supersolidity, “Related phases have been *observed in frustrated magnetic systems*²² or proposed to exist in the crust of neutron stars²³ and for excitons in semiconductor heterostructures²⁴”. We remark here our interest in noting the potential connections between these systems from completely different research areas, as a contribution to the spread of knowledge.

On page 3, it is claimed that a “paradigm shift” is made in this work, trying to boost the paper. Please remove this unnecessary and overdriven statement.

We have followed the suggestion of removing the “paradigm shift”. The sentence now reads: “In this work we demonstrate that it is possible to measure the superfluid fraction of a supersolid not only from...”.

Finally, I am wondering why a harmonic, elliptic confinement was chosen for this study, which brings along the additional complication that the supersolid droplet crystal is asymmetric, concerning the norm of the next-to center droplets and those close to the boundary. A better choice would be a box potential or, ideally, a ring confinement? All droplets would then be identical with $Z=1$.

To date, all available dipolar supersolids are produced under harmonic confinement and so they are inhomogeneous as in the case under study in our work.

Producing a homogeneous supersolid in a box-like trap is hindered by the peculiarity of the dipole-dipole interaction, which tends to produce an accumulation of the density in the edges. See the theory work: Supersolid edge and bulk phases of a dipolar quantum gas in a box, S. M. Roccuzzo, S. Stringari, and A. Recati, *Phys. Rev. Research* 4, 013086 (2022).

Therefore, one should go for a confinement intermediate between harmonic and box-like, which has been proposed theoretically but not yet realized experimentally. See:

How to realize a homogeneous dipolar Bose gas in the roton regime, Péter Juhász, Milan Krstajić, David Strachan, Edward Gandar, and Robert P. Smith, Phys. Rev. A 105 L061301 (2022).

Realizing a ring-shaped confinement is a very interesting option for the future. It would provide the same geometry of the original theoretical study by Leggett, although with the additional degree of freedom of the rotation of the clusters about their own axes, see the improved discussion before Fig. 4. Ring-shaped potentials are not yet available for dipolar supersolids. We are currently working towards implementing such a geometry, but it is a challenging task because of the need of realizing rings of microscopic size, just a few micrometres across, due to the high density and small size of the supersolid.

In conclusion, the study of dipolar supersolids in box-like or ring-shaped potentials is well beyond the current experimental capabilities, and we are bound to study harmonically trapped systems. With our current work, however, we demonstrated how it is possible to measure the superfluid fraction of a hypothetical homogeneous system whose unit cell is identical to the central cell of our inhomogeneous supersolid. Our results are therefore general and will apply to the future realizations of homogeneous dipolar supersolids, as well as to other supersolid candidates in condensed-matter systems that are intrinsically homogeneous.

A comparison is made between a Josephson self-induced junction lattice and a homogeneous superfluid, and it is concluded that the observed oscillations are clearly related to the formation of the intrinsic junction lattice. I wonder, has this been compared to the action of a periodic optical lattice in the background? How different are the oscillations in the latter case from the self-induced Josephson junction array?

For a standard superfluid where the density modulation is provided by a periodic optical lattice one has also a behaviour similar to that of a Josephson junction array. This is well studied in the field, see for example refs. 38-41 (original manuscript, now refs. 41-43), and we have personally contributed to those studies. In the presence of the fixed physical barriers provided by an optical lattice, the mapping to Josephson junction arrays is complete, because there aren't the additional degrees of freedom that appear in the supersolid, where instead the peaks and dips in the density can move. Also for superfluids in optical lattices one can define a superfluid fraction, see the discussion in the conclusions section of the manuscript and refs. 42-43 (original manuscript, now refs. 45-46). In that case there aren't however the novel phenomena predicted for supersolids, like the partially quantized vortices that the superfluid fraction would be useful to characterise, besides the Goldstone mode already discussed.

Therefore, we can say that superfluids in optical lattices are nowadays well understood and realise ordinary Josephson junction arrays, while supersolids realise a new type of array with additional degrees of freedom and new interesting phenomena that still need to be studied.

Some minor suggestions:

1) In the introduction you say that the superfluid fraction has not been assessed in the supersolid case. This is not generally true, as there are some theory works that precisely address this point. What is probably meant here is that this has not yet been assessed experimentally. Please change the text accordingly and give the proper references.

We have changed the text of the summary as suggested: “The superfluid fraction was introduced long ago, but until now its predicted sub-unity value has not been assessed experimentally in any supersolid candidate.” In our opinion, the original work by Leggett (ref. 16) remains the principal reference to the theoretical understanding of the superfluid fraction of supersolids, and most of the more recent theory works, such as the four papers we listed in the response to Referee #1, refer to it. Therefore, considering also the already large number of references in the bibliography, we would avoid introducing further references on this point.

2) On page 2, where it says “As shown in Fig. 1” it would be better to say “as sketched in Fig. 1” as the plot is schematic rather than containing any data.

We changed the text as suggested.

3) On page 3 the “individual cells of a supersolid” are discussed but it is not clear how these cells are defined? Please be more precise.

To clarify this point, we replaced the sentence “In Fig.1, one can note a similarity between individual cells of the supersolid and Josephson junctions, having the same structure of two bulk superfluids connected by a weak link.” with the sentence: “As sketched in Fig. 1, the unit cell of a 1D supersolid lattice is composed by two density maxima of the wavefunction connecting through a density minimum, so it has the typical structure of a Josephson junction, two bulk superfluids connected by a weak link.” This should clarify the definition of the unit cell of the supersolid lattice that we use in the rest of the manuscript.

4) On page 3, when the self-induced Josephson oscillation are discussed, please include again Ref. 29.

The Referee is referring to the sentence: “Here we demonstrate that a supersolid can in fact sustain coherent phase-density oscillations, behaving as an array of Josephson junctions.” Here we are referring to the general concept of a Josephson junction array, not to the specific model introduced by ref. 29 (originally, now ref. 20). We would therefore like to avoid citing ref. 29 in this sentence.

5) On page 5, when referring to a dipolar supersolid only reference 7 is cited. Please cite also 8 and 9, as you did at other places, for the discovery of supersolidity in dipolar systems.

In this part of the manuscript we are starting to describe our own system, which is very similar to the one we studied originally in ref. 7. That's why we were citing only ref. 7. We however added refs 8-9 as suggested, and cited ref. 7 again in the next sentence giving the details of our own system.

6) On page 5 you state that “interactions are weak, allowing accurate theoretical modeling”. This is only partially true, the applied extended Gross-Pitaevskii method is still only approximate. This should be mentioned.

We revised the text into “interactions are weak, allowing *relatively* accurate theoretical modeling³⁹”, adding the new ref. 39: Dilute dipolar quantum droplets beyond the extended Gross-Pitaevskii equation, F. Böttcher, et al., Phys. Rev. Research 1, 033088 (2019) that in our opinion best captures the limits of the extended mean-field theory normally used to describe dipolar supersolids.

7) On page 6 you speak about the “asymmetry between the left and right halves”. Please be more precise; asymmetry of what quantity? Norm?

On page 6, we speak about the “population imbalance Z between the left and right halves of the supersolid”. Later in the manuscript, Z is defined formally as “the population difference between the left and right halves of the system, $Z = (N_1 + N_2 + N_3 - N_4 - N_5 - N_6)/N$ ”. It seems to us that it is not possible to give the formal definition of Z right at the beginning, without making the whole discussion much heavier. To make more clear that the two Z s are the same quantity, we changed the text on page 6 to “population *difference* Z between the left and right halves of the supersolid”, so that it reads identically to the successive sentence. We hope that this modification clarifies the point.

8) I understood that U is proportional to the interaction energy, but where is it defined in the text? I got confused about this point when I came to Eq. 4b

We modified the sentence below eq. 4b, to explain what U is: “and U is the interaction energy per particle.”

9) Also, the discussion of the quantity Z and its definition should come earlier in the text.

Please see the response to point 7.

10) On page 7 you mention “transverse effects” but to the reader it is not obvious what these are; please explain.

Just above Fig. 4, the original manuscript contained the statement: "... the annular geometry originally studied by Leggett, where however our supersolid with macroscopic clusters would also show transverse effects not included in Leggett's theory for the moment of inertia. Studying the Josephson effect allows us to avoid such transverse effects." The main transverse effect we were implying is the possibility for the clusters to rotate about their own axes, perpendicular to the plane of a ring-shaped system under rotation. Leggett explicitly excluded this possibility by considering an infinitely thin ring. Using a linear system like we are doing in the present work, we can avoid such rotations.

We realised that this point is more appropriate in the conclusions, and that its discussion is more appropriate for the Methods. So, to improve the clarity on this point, we have added the following sentence to the conclusions: "The Josephson-oscillation technique works naturally in linear geometries and so it does not require any adaptation for the finite size of the clusters in the supersolid-like phases available in experiments ¹⁻¹², differently from the rotation technique ¹⁶ (see Methods)." And we have added the following paragraph to the Methods, section "Discussion of the Leggett model": "We note that in a linear system like the one employed in the experiment, the superfluid fraction measured from the Josephson dynamics is not affected by radial effects, which are instead relevant in the case of rotating systems ³¹. Indeed, the superfluid fraction extracted from a measurement of the moment of inertia, $I = (1 - f_s)I_c$, would also take into account the additional contribution given by the reduced inertia of the superfluid clusters composing the system, which rotate around their centers of mass. Leggett's upper bound is instead derived in the limit of an infinite radius of the annulus, where such radial effects can be neglected ¹⁶."

Additional changes

We corrected a couple of typos in the Methods section:

- "Standard deviation of the mean" instead of "standard deviation" in the section "Phase detection and analysis"
- A factor of two in equation for Z in the section "Six-mode Josephson model"

We added an acknowledgement of a funding source.

Sub-unity superfluid fraction of a supersolid from self-induced Josephson effect

G. Biagioni^{1,3,*}, N. Antolini^{2,3,*}, B. Donelli^{2,4,6,8}, L. Pezzè^{2,4,6}, A. Smerzi^{2,4,6}, M. Fattori^{1,2,5}, A. Fioretti³, C. Gabbanini³, M. Inguscio^{2,7}, L. Tanzi^{2,3}, G. Modugno^{1,2,3}

¹ Dipartimento di Fisica e Astronomia, Università degli studi di Firenze, Via G. Sansone 1, 50019 Sesto Fiorentino, Italy

² European Laboratory for Nonlinear Spectroscopy, Università degli studi di Firenze, Via N. Carrara 1, 50019 Sesto Fiorentino, Italy

³ CNR-INO, sede di Pisa, Via Moruzzi 1, 56124 Pisa, Italy

⁴ CNR-INO, sede di Firenze, Largo Enrico Fermi 5, 50125 Firenze, Italy

⁵ CNR-INO, sede di Sesto Fiorentino, Via N. Carrara 1, 50019 Sesto Fiorentino, Italy

⁶ QSTAR, Largo Enrico Fermi 2, 50125 Firenze, Italy

⁷ Dipartimento di Ingegneria, Università Campus Bio-Medico di Roma, Via Alvaro Del Portillo 21, 00128 Roma, Italy

⁸ Università degli studi di Napoli "Federico II", Via Cinthia 21, 80126 Napoli, Italy

*These authors contributed equally

Recently, a novel class of superfluids and superconductors with spontaneous spatially-periodic modulation of the superfluid density has been discovered in a variety of systems¹⁻¹². They might all be related to the concept of a supersolid phase of matter, which has by definition a macroscopic wavefunction with spatial modulation originating from the simultaneous, spontaneous breaking of the gauge and translational symmetries¹³⁻¹⁶. However, this relation has only been recognized in some cases^{1,2,5-9} and there is the need for universal properties quantifying the differences between supersolids and ordinary superfluids/superconductors or ordinary crystals. A key property is the superfluid fraction¹⁶, which measures the reduction in superfluid stiffness due to the spatial modulation, leading to the non-standard superfluid dynamics of supersolids¹⁶⁻¹⁸. The superfluid fraction was introduced long ago, but until now its predicted sub-unity value has not been quantitatively assessed experimentally in any supersolid candidate. Here we propose and demonstrate an innovative method to measure the superfluid fraction of a supersolid based on the Josephson effect, a phenomenon ubiquitous in superfluids and superconductors, associated with the presence of a physical barrier between two superfluids¹⁹. ~~We show that in a supersolid the Josephson effect arises spontaneously, without the need of a barrier, due to the spontaneous spatial modulation. Individual lattice cells of the supersolid lattice behave as self-induced Josephson junctions.~~ **A supersolid can be modelled as an array of Josephson junctions²⁰, due to the spontaneous spatial modulation of the wavefunction. Here**

we demonstrate, through direct measurements, that individual cells of a supersolid can sustain Josephson oscillations, and we show that from the current-phase dynamics of the Josephson oscillations we can derive directly the local superfluid fraction. Our investigation is carried out on a cold-atom dipolar supersolid⁷, for which we discover a relatively large sub-unity superfluid fraction that makes realistic the study of novel phenomena such as partially quantized vortices and supercurrents¹⁶⁻¹⁸. Our methods and results are very general and open a new direction of research that may unify the description of all supersolid-like systems. Our work also discloses a new type of Josephson junction.

Supersolids are a fundamental phase of matter originated by the spontaneous breaking of the gauge symmetry as in superfluids and superconductors, and of the translational symmetry as in crystals¹³⁻¹⁶. This gives rise to a macroscopic wavefunction with spatially-periodic modulation, and to mixed superfluid and crystalline properties. Supersolids were originally predicted in the context of solid helium¹³⁻¹⁶. Today, quantum phases with spontaneous modulation of the wavefunction are under study in a variety of bosonic and fermionic systems. These include: the second layer of ⁴He on graphite^{1,2}; ultracold quantum gases in optical cavities⁵, with spin-orbit coupling⁶ or with strong dipolar interactions^{7-9,21}; the pair density wave phase of ³He under confinement^{3,4}; pair density wave phases in various types of superconductors¹⁰⁻¹². Related phases have been observed in frustrated magnetic systems²² or proposed to exist in the crust of neutron stars²³ and for excitons in semiconductor heterostructures²⁴. All these systems might be connected to supersolidity. The periodic structure of the wavefunction of all these systems is a prerequisite for supersolidity, which however so far has emerged clearly only in some cold-atom systems with the evidence of the double spontaneous symmetry breaking and of the mixed superfluid-crystalline character^{25,26,5}. The experiments carried out so far on the other types of systems have proved the coexistence of superfluidity/superconductivity and crystal-like structure^{1-4,10-12}, but no quantitative connection of the observations to the concept of supersolidity has been made. One of the difficulties in comparing different types of systems with spatial modulation of the wavefunction is the seeming lack of a universal property quantifying the deviations from the dynamical behaviour of ordinary superfluids or superconductors.

Here we note that a property with such characteristics already exists, the so-called superfluid fraction of supersolids, well known in the field of superfluids but not in the one of superconductors. The superfluid fraction, introduced by A. J. Leggett in 1970¹⁶, quantifies the effect of the spatial modulation on the superfluid stiffness, which is in itself a defining property of superfluids and superconductors. The superfluid stiffness indeed measures the finite energy cost of twisting the phase of the macroscopic wavefunction and accounts for all fundamental phenomena of superfluidity, such as phase coherence, quantized vortices and supercurrents²⁷. As shown sketched in Fig.1, while in a homogeneous superfluid/superconductor the phase varies linearly in space, in a modulated system most of the phase variation can be accommodated in the minima of the density, reducing the energy cost. Since the superfluid velocity is the gradient of the phase, this implies that peaks and valleys should move differently, giving rise to complex dynamics with mixed classical (crystalline) and quantum (superfluid) character. For example, fundamental superfluid phenomena like vortices and supercurrents are predicted to be profoundly affected by the

presence of the spatial modulation, losing the canonical quantization of their angular momentum^{16-18,28}. The superfluid fraction, which ranges from unity for standard superfluids to zero for standard crystals, enters directly in all these phenomena and is therefore the proper quantity to assess the deviations from standard superfluids and superconductors. Note that the superfluid fraction of supersolids is not related to thermal effects, in contrast to the superfluid fraction due to the thermal depletion of superfluids and superconductors²⁹.

The standard methods to measure the superfluid stiffness are based on the measurement of global properties such as the moment of inertia for rotating superfluids^{1,2}, or the penetration depth of the magnetic field for superconductors³⁰. In dipolar supersolids, previous attempts using rotational techniques revealed a large superfluid fraction³¹, but were not precise enough to assess its sub-unity value^{32,33}. In the other systems there is evidence that the superfluid stiffness is low^{1,2,30}, but no quantitative measurement of a sub-unity superfluid fraction is available.

In this work ~~we make a paradigm shift, demonstrating~~ **we demonstrate** that it is possible to measure the superfluid fraction **of a supersolid** not only from global dynamics but also from a fundamental phenomenon taking place in individual cells of **a the supersolid lattice**: the Josephson effect¹⁹. ~~In Fig. 1, one can note a similarity between individual cells of the supersolid and Josephson junctions, having the same structure of two bulk superfluids connected by a weak link.~~ **As sketched in Fig. 1, the unit cell of a 1D supersolid lattice is composed by two density maxima connecting through a density minimum, so it has the typical structure of a Josephson junction, two bulk superfluids connected by a weak link.** It is therefore tempting to associate supersolidity to the very existence of local Josephson dynamics. ~~However, so far the Josephson effect in supersolids has not been studied, beyond the phenomenological modelling of the relaxation towards the ground state in a cold atom system²⁹. There is no theoretical or experimental evidence for local Josephson oscillations, nor an understanding of the potential relation between the Josephson effect and the superfluid stiffness.~~ **So far, the analogy between a supersolid and an array of Josephson junctions has been only employed to model the relaxation towards the ground state of a dipolar supersolid²⁰. There is instead no theoretical or experimental evidence for local Josephson oscillations, nor an understanding of the potential relation between the Josephson effect and the superfluid fraction.** The problem is complicated by the fact that, in supersolids, the weak links are self-induced by internal interactions rather than by an external potential, so they can change during the dynamics. Therefore it is not clear if phenomena such as Josephson oscillations can exist at all in a supersolid.

Here we demonstrate **experimentally and theoretically** that a supersolid can in fact sustain coherent phase-density oscillations, behaving as an array of Josephson junctions. We also show that the Josephson coupling energy that one can deduce from the Josephson oscillations provides a direct measurement of the ~~local~~ superfluid fraction. We use this novel approach to measure with high precision the superfluid fraction of the dipolar supersolid appearing in a quantum gas of magnetic atoms. We find a whole range of sub-unity values of the superfluid fraction, depending on the depth of the density modulation in accordance with Leggett's predictions.

FIG. 1 Sketch of the superfluid fraction from the application of a phase twist in a bosonic system at zero temperature. a) In a homogeneous superfluid a phase twist with amplitude $\Delta\varphi$ results in a constant gradient of the phase, i.e. a constant velocity, while in a supersolid (b,c) the kinetic energy can be minimised by accumulating most of the phase variation in the low-density regions. The grey and green areas represent the number density and the kinetic energy density respectively, while the phase profile is plotted in red. The superfluid fraction is the ratio of the area under the green curve to that of the homogeneous case. (b) Leggett's approach, which for an annular system would correspond to a stationary rotation, leads to a monotonous increase of the phase. (c) Our method, based on an alternating oscillation of the phase, leads to Josephson oscillations. Both kinetic energy and superfluid fraction are the same for b) and c).

Leggett's approach to the superfluid fraction considers an annular supersolid in the rotating frame and maps it to a linear system with an overall phase twist, as sketched in Fig. 1b. The superfluid fraction is defined on a unit cell as^{16,34}

$$f_s = \frac{E_{kin}}{E_{kin}^{hom}}. \quad (1)$$

The numerator is the kinetic energy acquired by the supersolid with number density $n(x)$ when applying a phase twist $\Delta\varphi$ over a lattice cell of length d ,

$$E_{kin} = \frac{\hbar^2}{2m} \int_{cell} dx n(x) |\nabla\varphi(x)|^2, \text{ and thus accounts for density and phase modulations.}$$

The denominator $E_{kin}^{hom} = Nm v_s^2/2$ is the kinetic energy of a homogeneous superfluid of N atoms and velocity $v_s = \hbar\Delta\varphi/(md)$ associated with a constant phase gradient $\Delta\varphi$ across

the cell. Using a variational approach ^{16,35}, Leggett found an upper and a lower bound for Eq. (1), $f_s^l \leq f_s \leq f_s^u$, see Methods. In particular, the upper bound

$$f_s^u = \left(\frac{1}{d} \int_0^d \frac{dx}{\bar{n}(x)} \right)^{-1}, \quad (2)$$

where $\bar{n}(x)$ is the normalised 1D density, restricts f_s to be lower than unity if the density is spatially modulated. **Note that the calculation of the superfluid fraction, which is a global property, by considering a single lattice cell is possible due to the periodicity of the wavefunction of the supersolid ¹⁶.**

We propose an alternative expression for the superfluid fraction, considering Josephson phase twists with alternating sign between neighbouring lattice sites of a supersolid, as sketched in Fig. 1c. This corresponds to a different type of motion of the supersolid, with no global flow but with alternate Josephson phase-density oscillations between sites. Also in this case we can consider a single cell, since the kinetic energy is proportional to $|\nabla\varphi(x)|^2$, so it does not depend on the sign of the phase twist. In the limit of small excitations ($\Delta\varphi \rightarrow 0$), the kinetic energy of a Josephson junction is given by $E_{kin} = NK\Delta\varphi^2$, where K is the coupling energy across the barrier ³⁶. From Eq. (1) we thus find:

$$f_s = \frac{K}{\hbar^2/(2md^2)}, \quad (3)$$

showing a direct relation between the superfluid fraction and the coupling energy of the junction. We note that an expression similar to the upper bound in Eq. (2) was derived by Leggett for the coupling energy of a single Josephson junction ³⁷, however without discussing the connection to the superfluid fraction.

We now demonstrate the existence of coherent Josephson-like oscillations in a dipolar supersolid ⁷⁻⁹. This system is particularly appealing to study fundamental aspects of supersolidity ³⁸: the supersolid lattice is macroscopic, with many atoms per site and large superfluid effects; the available control of the quantum phase transition allows to directly compare supersolids and superfluids; interactions are weak, allowing **a fairly** accurate theoretical modelling ³⁹. Our experimental system ⁷ is composed of about $N = 3 \times 10^4$ bosonic dysprosium atoms, held in a harmonic trap elongated along the x direction, with trap frequencies $(\omega_x, \omega_y, \omega_z) = 2\pi(18, 97, 102)$ Hz. By tuning the relative strength ϵ_{dd} of dipolar and contact interactions, we can cross the quantum phase transition from a standard Bose-Einstein condensate to the supersolid regime (Methods). The supersolid lattice structure is one dimensional, leading to a continuous phase transition ⁴⁰. Our typical supersolid is made of two main central clusters and four smaller lateral ones, with a lattice period $d \approx 4 \mu\text{m}$, as shown in Figure 2. We can vary the density modulation depth by varying the interaction strength in the range $\epsilon_{dd} = 1.38 - 1.45$; further increasing ϵ_{dd} leads to the

formation of an incoherent crystal of separate clusters, the so-called droplet crystal, a regime that we cannot study experimentally due to its short lifetime⁷.

Since our system is inhomogeneous, we focus our attention on the central cell, the one delimited by the clusters 3 and 4 in Fig. 2. As we will show, the superfluid fraction we derive from that cell corresponds to the superfluid fraction of a hypothetical homogeneous supersolid with all cells identical to the central one, as in Fig. 1, which is the system of general interest.

FIG. 2 - Josephson oscillations in a supersolid. (a) Sketch of the experimental system. The black line is the supersolid density profile at equilibrium. The green dashed line is the optical lattice potential used for the phase imprinting. (b) Examples of experimental single shots and corresponding integrated 1D profiles. Top row: interference fringes after a free expansion. Red curves are fit functions used to extract the phase difference $\Delta\phi$. Bottom row: in-situ images. Shaded areas indicate the populations of the left and right halves of the supersolid used to extract the population imbalance Z . (c) Oscillations of Z as a function of time at $\epsilon_{\text{od}} = 1.428$. Dots are experimental points. **Error bars are the standard deviation of the mean of about 20-30 measurements.** The solid line is the numerical simulation for the same parameters. The dotted line is a sinusoidal fit to the experimental data. (d) Same for $\Delta\phi$. **Experimental values and error bars are calculated using the circular mean and standard deviation of the mean (see Methods).**

We find that the application for a short time of an optical lattice with twice the spacing of the supersolid (sketched in Fig. 2a) imprints the proper alternating phase difference between adjacent clusters to excite Josephson oscillations. With a depth of 100 nK and an application time of $\tau=100 \mu\text{s}$ we obtain a phase difference of the order of $\pi/2$. After a variable evolution time in the absence of the lattice, we measure both the evolving phase difference $\Delta\varphi$ between neighbouring clusters and the population imbalance difference Z between the left and right halves of the supersolid. $\Delta\varphi$ is measured from the interference fringes developing after a free expansion (snapshots in Fig. 2b, top row), while Z is measured by in-situ phase-contrast imaging (Fig. 2b, bottom row) (Methods). As shown in Fig. 2c-d, we observe single-frequency oscillations of Z and $\Delta\varphi$, with the characteristic $\pi/2$ phase shift of the standard Josephson dynamics^{19,36,41-44}. The observation time is limited to about 100 ms by the finite lifetime of the supersolid, due to unavoidable particle losses⁷. The experimental observations agree very well with numerical simulations based on the time-dependent extended Gross-Pitaevskii equation, also shown in Fig. 2c-d (Methods). We have checked that the Josephson oscillations are not observable if we apply the same procedure to standard Bose-Einstein condensates, instead of supersolids (see Methods).

The observation of a single frequency in both experiment and simulations indicates not only that it is possible to excite Josephson-like oscillations in a supersolid, but also that they are a normal mode of the system. To model our observations, we develop a six-mode model, generalising the two-mode Josephson oscillations³⁶ to the case of six clusters (see Methods). We associate to the j th cluster a population N_j and a phase φ_j ($j = 1, \dots, 6$). In general, the dynamics includes contributions from each cluster and shows multiple frequencies. However, we find that, under appropriate conditions among the interaction and coupling energies, there exists a normal mode of the system where the dynamical variables of the two central clusters of the supersolid decouple from the lateral ones. This results in Josephson-like oscillations described by the two coupled equations

$$\dot{\Delta N} = -4KN_{34} \sin(\Delta\varphi), \quad (4a)$$

$$\dot{\Delta\varphi} = U\Delta N, \quad (4b)$$

where $\Delta N = N_3 - N_4$, $N_{34} = N_3(0) + N_4(0)$, and $\Delta\varphi = \varphi_3 - \varphi_4$, and U is the interaction energy per particle. These equations hold for interaction energies $N_{34}U$ much larger than K (for our system, $N_{34}U/(2K) > 25$, see Methods). Since in our case $\Delta N \ll N$, we keep only linear terms in $\Delta N/N$.

Equations 4(a-b) are equivalent to those of a simple pendulum with angle $\Delta\varphi$ and angular momentum ΔN , and in the small-angle limit feature sinusoidal oscillations with a single frequency, $\omega_j = \sqrt{4KUN_{34}}$. We emphasise that the current-phase relation Eq.(4a) as well as ω_j^2 differ by a factor 2 with respect to the Josephson equations of two weakly coupled Bose-Einstein condensates, due to the contribution of the lateral clusters, but are equal to the ones of a hypothetical homogeneous supersolid. Notice also that Eqs. 4(a-b) depend

only on the coupling energy K and the interaction energy U of the two central clusters, in contrast to the expectation that the inhomogeneity of the trapped system may introduce other energies in the equations of motion. We checked by Gross-Pitaevskii simulations that our experimental configuration satisfies the conditions to have a Josephson-like normal mode (namely Eq. 7 of the Methods).

In the experiment we are not able to resolve the population of the individual clusters, but we study the population difference between the left and right halves of the system, $Z = (N_1 + N_2 + N_3 - N_4 - N_5 - N_6)/N$. There is a proportionality relation between the two observables, $\Delta N = 2NZ$, which allows us to rewrite Eqs. 4(a-b) in terms of the experimental observables (Methods).

An important difference between a cell of the supersolid and a standard Josephson junction is the fact that in the supersolid the position of the weak link is not fixed by an external barrier but it is self-induced, so it can move. This leads to the appearance of a low-energy Goldstone mode associated with the spontaneous translational symmetry breaking. In a harmonic potential, it consists of a slow oscillation of the position of the weak link, together with the density maxima, and an associated oscillation of both Z and $\Delta\varphi$ ²⁶. Due to its low frequency (few Hz), the Goldstone mode is spontaneously excited by thermal fluctuations, resulting in shot-to-shot fluctuations of the experimental observables. The same low frequency, however, allows to separate Josephson and Goldstone dynamics in both experiment and theory (Methods).

We measure the Josephson frequency ω_j from a sinusoidal fit of the phase and population dynamics in Fig. 2c-d. We repeat the measurement by varying the interaction parameter ε_{dd} , corresponding to different depths of the supersolid density modulation. Figure 3 shows the fitted frequencies as a function of ε_{dd} and a comparison with the numerical simulations. We observe a decrease of the frequency for increasing ε_{dd} . This is justified by the fact that the superfluid current across the junction decreases because a larger and larger portion of the wavefunction remains localised inside the clusters, see insets in Fig. 3. This reduces the coupling energy K while only weakly affecting the interaction energy.

FIG.3 - Josephson oscillation frequencies as a function of the interaction parameter ϵ_{dd} . Red dots are the experimental frequencies for $\Delta\varphi$. Filled and open blue dots are the frequencies for Z measured by in situ imaging with and without optical separation, respectively (Methods). Vertical error bars are the uncertainties in the non-linear fit of the sinusoidal oscillations displayed in Fig.2c-d. Horizontal error bars represent our experimental resolution in ϵ_{dd} (Methods). The red point at $\epsilon_{dd} = 1.444$ is slightly shifted horizontally for clarity. Black points are the results of numerical simulations. The dashed line is a guide for the eye. The insets show the modulated ground state density profiles obtained from numerical simulations for different values of ϵ_{dd} . The vertical dotted line marks the critical point of the superfluid-supersolid quantum phase transition.

From the Josephson frequency we can derive the coupling energy as $K = \omega_J^2 / (4UN_{34})$, with the denominator obtained from the simulations. We verified that this relation holds not only in the small-amplitude regime of the simulations, but also for the larger amplitudes of the experiment.

From the measured K , we derive in turn the superfluid fraction using Eq. 3. The results are shown in Figure 4 and feature a progressive reduction of the superfluid fraction below unity for increasing depths of the supersolid modulation. The experimental data are in good agreement with the numerical simulations (green dots), where according to Eq. (4a) the coupling energy is obtained from the linear dependence of dZ/dt on $\sin(\Delta\varphi)$ (current-phase relation), see Fig. 4b. A similar analysis (Fig. 4c) was performed on the experimental data for which we have combined phase and population oscillations (pink dots in Fig. 4a). The results for these data points demonstrate the reduced superfluid fraction of the supersolid with no numerical input on the interaction energy U .

~~What we measure here is the superfluid fraction of the central cell of our inhomogeneous supersolid, for a one-dimensional phase twist. Such a local quantity would coincide with the global superfluid fraction of a hypothetical homogeneous supersolid, composed of cells identical to our central one. This includes the annular geometry originally studied by Leggett, where however our supersolid with macroscopic clusters would also show transverse~~

effects, not included in Leggett's theory for the moment of inertia⁴⁶. Studying the Josephson effect allows us to avoid such transverse effects.

FIG. 4 - Superfluid fraction of the supersolid from Josephson oscillations. (a) Superfluid fraction as a function of ϵ_{dd} . Black dots are experimental results derived from the Josephson frequencies. Vertical error bars result from the error propagation of Eq.3, with $K = \omega_J^2 / (4UN_{34})$, see Methods. Green dots are results from numerical simulations. Error bars are the uncertainties of the linear fits employed to determine K and UN_{34} . Pink points are derived from the experimental phase-current relation, as in (c). Error bars are estimated using the propagation of Eq.3, with K and its relative uncertainty extracted from linear fits of experimental data. The open pink point at $\epsilon_{dd} = 1.444$ is the dataset without the optical separation technique (Methods). The grey band spans from the upper to the lower bound of Eq. (3). (b-c) Phase-current relation at $\epsilon_{dd} = 1.444$. The points show the results of numerical simulations (b) and experimental measurements (c), respectively. From the linear regressions (green and pink lines) we extract the coupling energy K according to Eq. (4a 5a). The shaded regions are the confidence bands for one standard deviation.

In Fig. 4a we also compare our results with Leggett's prediction of Eq. (2), relating the superfluid fraction to the density modulation of the supersolid. From the numerical density profiles, we calculate both the upper bound f_s^u and the corresponding lower bound³⁵ f_s^l , which delimit the grey area in Fig. 4a (see Methods). The two bounds would coincide if the density distribution were separable in the transverse coordinates y and z . Since our supersolid lattice is one dimensional, the two bounds are close to each other. The superfluid

fraction calculated from the simulated dynamics lies between the two bounds in the whole supersolid region we investigated, demonstrating the applicability of Leggett's result to our system.

In conclusion, the overall agreement between experiment, simulations and theory on our dipolar supersolid proves the long-sought sub-unity superfluid fraction of supersolids and its relation to the spatial modulation of the superfluid density. The demonstration of self-sustained Josephson oscillations in a supersolid ~~not only establishes an analogy between supersolids and Josephson junction arrays, but also~~ provides a novel proof of the extraordinary nature of supersolids compared to ordinary superfluids and crystals. These oscillations indeed cannot exist neither in crystals, where particles are bound to lattice sites, nor in ordinary superfluids, which do not have a lattice structure.

Our findings open new research directions. The observed reduction of the superfluid fraction with increasing modulation depths may explain the low superfluid stiffness measured in other systems, such as ^4He on graphite ^{1,2} or superconductors hosting pair-density-wave phases ¹⁰⁻¹². An important question related to the pair density waves in fermionic systems is how Leggett's bounds on the superfluid fraction may be extended to systems where the superfluid density and particle density do not coincide. Note that Eq. (2) is also applicable to standard superfluids with an externally-imposed spatial modulation, as demonstrated for Bose-Einstein condensates in optical lattices via measurements of the effective mass ⁴¹ or of the sound velocity ^{45,46}. In the supersolid, however, the dynamics linked to the reduced superfluid fraction is not constrained by an external potential, and so totally new phenomena might be observed. The large value of f_s we measured for the dipolar supersolid, which remains larger than 10% also for deep density modulations, indicates that partially quantized supercurrents ^{16,18} and vortices ¹⁷ should appear at a macroscopic level.

Due to the generality of the Josephson effect, our Josephson-oscillation technique might be applied to characterise the local superfluid dynamics of the other supersolid-like phases under study in superfluid and superconducting systems. Eq. 3 is applicable in general, considering that the detection of Josephson oscillations implies measurement of both the coupling energy and the spatial period of the superfluid density modulation. For example, a promising type of system may be the pair-density wave phase in superconductors, where the modulation has already been resolved. **The Josephson-oscillation technique works naturally in linear geometries and so it does not require any adaptation for the finite size of the clusters in the supersolid-like phases available in experiments ¹⁻¹², differently from the rotation technique ¹⁶ (see Methods).**

Additionally, the self-induced Josephson junctions we have identified in supersolids might have extraordinary properties due to the mobility of the weak links. Indeed, although the Goldstone mode of the weak links is not relevant for the Josephson dynamics due to its very low energy, for the same reason it may affect the fluctuation properties of the junction ⁴⁷, potentially leading to new thermometry methods ⁴⁸, and especially to novel entanglement properties ⁴⁹.

Bibliography

1. J. Nyéki, A. Phillis, A. Ho, D. Lee, P. Coleman, J. Parpia, B. Cowan, and J. Saunders, Intertwined superfluid and density wave order in two-dimensional ^4He , *Nat. Phys.* 13, 455-459 (2017).
2. J. Choi, A. A. Zadorozhko, J. Choi, and E. Kim, Spatially modulated superfluid state in two-dimensional ^4He films, *Phys. Rev. Lett.* 127, 135301 (2021).
3. L. V. Levitin, B. Yager, L. Sumner, B. Cowan, A. J. Casey, J. Saunders, N Zhelev, R. G. Bennett, and J. M. Parpia, Evidence for a spatially modulated superfluid phase of ^3He under confinement, *Phys. Rev. Lett.* 122, 085301 (2019).
4. A. J. Shook, V. Vadakkumbatt, P. Senarath Yapa, C. Doolin, R. Boyack, P. H. Kim, G. G. Popowich, F. Souris, H. Christani, J. Maciejko, and J. P. Davis, Stabilized pair density wave via nanoscale confinement of superfluid ^3He , *Phys. Rev. Lett.* 124, 015301 (2020).
5. J. Léonard, A. Morales, P. Zupancic, T. Esslinger, and T. Donner. Supersolid formation in a quantum gas breaking a continuous translational symmetry, *Nature* 543, 87–90 (2017).
6. J.-R. Li, J. Lee, W. Huang, S. Burchesky, B. Shteynas, F. Ç. Top, A. O. Jamison and W. Ketterle, A stripe phase with supersolid properties in spin–orbit-coupled Bose–Einstein condensates, *Nature* 543, 91–94 (2017).
7. L. Tanzi, E. Lucioni, F. Famà, J. Catani, A. Fioretti, C. Gabbanini, R. N. Bisset, L. Santos, and G. Modugno, Observation of a dipolar quantum gas with metastable supersolid properties, *Phys. Rev. Lett.* 122, 130405 (2019).
8. F. Böttcher, J. Schmidt, M. Wenzel, J. Hertkorn, M. Guo, T. Langen, and T. Pfau, Transient supersolid properties in an array of dipolar quantum droplets, *Phys. Rev. X* 9, 011051 (2019).
9. L. Chomaz, D. Petter, P. Ilzhöfer, G. Natale, A. Trautmann, C. Politi, G. Durastante, R. M. W. van Bijnen, A. Patscheider, M. Sohmen, M. J. Mark, and F. Ferlaino, Long-lived and transient supersolid behaviours in dipolar quantum gases, *Phys. Rev. X*, 9, 021012 (2019).
10. M. H. Hamidian, S. D. Edkins, S. H. Joo, A. Kostin, H. Eisaki, S. Uchida, M. J. Lawler, E.-A. Kim, A. P. Mackenzie, K. Fujita, J. Lee, J. C. S. Davis, Detection of a Cooper-pair density wave in $\text{Bi}_2\text{Sr}_2\text{CaCu}_2\text{O}_{8+x}$. *Nature* 532, 343–347 (2016).
11. Y. Liu, T. Wei, G. He, Y. Zhang, Z. Wang and J. Wang, Pair density wave state in a monolayer high- T_c iron-based superconductor, *Nature* 618, 934–939 (2023).
12. D. F. Agterberg, J.C. S. Davis, S. D. Edkins, E. Fradkin, D. J. Van Harlingen, S. A. Kivelson, P. A. Lee, L. Radzihovsky, J. M. Tranquada, and Y. Wang, The physics of pair density waves: cuprate superconductors and beyond, *Annual Review of Condensed Matter Physics* 11, 231 (2020).

13. E. P. Gross, Unified theory of interacting bosons, *Phys. Rev.* 106, 161 (1957).
14. A. F. Andreev and I. M. Lifshitz, Quantum theory of defects in crystals, *Sov. Phys. JETP* 29, 1107–1113 (1969).
15. G. V. Chester, Speculations on Bose–Einstein condensation and quantum crystals, *Phys. Rev. A* 2, 256–258 (1970).
16. A. J. Leggett, Can a solid be superfluid?, *Phys. Rev. Lett.* 25, 1543 (1970).
17. A. Gallemì, S.M. Roccuzzo, S. Stringari and A. Recati, Quantized vortices in dipolar supersolid Bose-Einstein-condensed gases, *Phys. Rev. Lett. A* 102, 023322 (2020).
18. M. Nilsson Tengstrand, D. Bohlm, R. Sachdeva, J. Bengtsson, and S. M. Reimann, Persistent currents in toroidal dipolar supersolids, *Phys. Rev. A* 103, 013313 (2021).
19. B. D. Josephson, Possible new effects in superconductive tunnelling, *Phys. Lett.* 1, 251–253 (1962).
20. P. Ilzhöfer, M. Sohmen, G. Durastante, C. Politi, A. Trautmann, G. Natale, G. Morpurgo, T. Giamarchi, L. Chomaz, M. J. Mark and F. Ferlaino, Phase coherence in out-of-equilibrium supersolid states of ultracold dipolar atoms, *Nat. Phys.* 17, 356–361 (2021).
21. M.A. Norcia, C. Politi, L. Klaus, M. Sohmen, M. J. Mark, R. N. Bisset, L. Santos and F. Ferlaino, Two-dimensional supersolidity in a dipolar quantum gas. *Nature* 596, 357–361 (2021).
22. J. Xiang et al. Giant magnetocaloric effect in spin supersolid candidate $\text{Na}_2\text{BaCo}(\text{PO}_4)_2$, *Nature* 625, pages 270–275 (2024).
23. C. J. Pethick, N. Chamel and S. Reddy, Superfluid dynamics in neutron star crusts, *Progress of Theoretical Physics Supplement* 186, 9-16 (2010).
24. S. Conti, A. Perali, A. R. Hamilton, M. V. Milošević, F. M. Peeters, and D. Neilson, Chester supersolid of spatially indirect excitons in double-layer semiconductor heterostructures, *Phys. Rev. Lett.* 130, 057001 (2023).
25. L. Tanzi, S. M. Roccuzzo, E. Lucioni, F. Famà, A. Fioretti, C. Gabbanini, G. Modugno, A. Recati, and S. Stringari, Supersolid symmetry breaking from compressional oscillations in a dipolar quantum gas, *Nature* 574, 382 (2019).
26. Guo, M., Böttcher, F., Hertkorn, J.-N. Schmidt, M. Wenzel, H. P. Büchler, T. Langen and T. Pfau, The low-energy Goldstone mode in a trapped dipolar supersolid, *Nature* 574, 386–389 (2019)
27. A.J. Leggett, Superfluidity, *Rev. Mod. Phys.* 71, S318 (1999).
28. G. Biagioni, Evidence of superfluidity in a dipolar supersolid, *Il Nuovo Cimento* 44 C, 121 (2021).

29. L. Landau, Theory of the superfluidity of helium II. *Phys. Rev.* Vol. 60, 356–358 (1941).
30. I. Božović, X. He, J. Wu and A.T. Bollinger, Dependence of the critical temperature in overdoped copper oxides on superfluid density. *Nature* 536, 309–311 (2016).
31. L. Tanzi, J. G. Maloberti, G. Biagioni, A. Fioretti, C. Gabbanini, and G. Modugno, Evidence of superfluidity in a dipolar supersolid from non-classical rotational inertia, *Science* 371, 1162 (2021).
32. M. A. Norcia, E. Poli, C. Politi, L. Klaus, T. Bland, M. J. Mark, L. Santos, R. N. Bisset, and F. Ferlaino, Can angular oscillations probe superfluidity in dipolar supersolids?, *Phys. Rev. Lett.* 129, 040403 (2022).
33. S. M. Roccuzzo, A. Recati, and S. Stringari, Moment of inertia and dynamical rotational response of a supersolid dipolar gas, *Phys. Rev. A* 105, 023316 (2022).
34. M. E. Fisher, M. N., Barber and D. Jasnow, Helicity modulus, superfluidity, and scaling in isotropic systems, *Phys. Rev. A* 8, 1111 (1973).
35. A. J. Leggett, On the superfluid fraction of an arbitrary many-body system at $T=0$, *J. Stat. Phys.* 93, 927 (1998).
36. A. Smerzi, S. Fantoni, S. Giovanazzi, and S. R. Shenoy, Quantum coherent atomic tunneling between two trapped Bose-Einstein condensates, *Phys. Rev. Lett.* 79, 4950 (1997).
37. I. Zapata, F. Sols and A. J. Leggett, Josephson effect between trapped Bose-Einstein condensates, *Phys. Rev. A* 57, R28 (1998).
38. L. Chomaz, I. Ferrier-Barbut, F. Ferlaino, B. Laburthe-Tolra, B. L. Lev and T. Pfau, Dipolar physics: a review of experiments with magnetic quantum gases, *Rep. Prog. Phys.* 86 026401 (2022).
39. F. Böttcher, M. Wenzel, J.-N. Schmidt, M. Guo, T. Langen, I. Ferrier-Barbut, T. Pfau, R. Bombín, J. Sánchez-Baena, J. Boronat, and F. Mazzanti, Dilute dipolar quantum droplets beyond the extended Gross-Pitaevskii equation, *Phys. Rev. Research* 1, 033088 (2019).
40. G. Biagioni, N. Antolini, A. Alaña, M. Modugno, A. Fioretti, C. Gabbanini, L. Tanzi, and G. Modugno, Dimensional crossover in the superfluid-supersolid quantum phase transition, *Phys. Rev. X* 12, 021019 (2022).
41. F. S. Cataliotti, S. Burger, C. Fort, P. Maddaloni, F. Minardi, A. Trombettoni, A. Smerzi, and M. Inguscio, Josephson junction arrays with Bose-Einstein condensates, *Science*, 293, 5531 (2001).
42. M. Albiez, R. Gati, J. Fölling, S. Hunsmann, M. Cristiani, and M. K. Oberthaler, Direct observation of tunneling and nonlinear self-trapping in a single bosonic Josephson junction, *Phys. Rev. Lett.* 95, 010402 (2005).

43. S. Levy, E. Lahoud, I. Shomroni and J. Steinhauer, The a.c. and d.c. Josephson effects in a Bose–Einstein condensate, *Nature* 449, 579–583 (2007).
44. G. Valtolina, A. Burchianti, A. Amico, E. Neri, K. Xhani, J. A. Seman, A. Trombettoni, A. Smerzi, M. Zaccanti, M. Inguscio and G. Roati, Josephson effect in fermionic superfluids across the BEC-BCS crossover, *Science*, 350, 6267 (2015).
45. J. Tao, M. Zhao and I. B. Spielman, Observation of anisotropic superfluid density in an artificial crystal, arXiv:2301.01258
46. G. Chauveau, C. Maury, F. Rabec, C. Heintze, G. Brochier, S. Nascimbene, J. Dalibard, J. Beugnon, S. M. Roccuzzo and S. Stringari, Superfluid fraction in an interacting spatially modulated Bose-Einstein condensate, *Phys. Rev. Lett.* 130 226003 (2023).
47. T. Berrada, S. van Frank, R. Bücke, T. Schumm, J.-F. Schaff, and J. Schmiedmayer, Integrated Mach-Zehnder interferometer for Bose-Einstein condensates, *Nat. Comm.* 4, 2077 (2013).
48. R. Gati, B. Hemmerling, J. Foëlling, M. Albiez, and M. K. Oberthaler, Noise thermometry with two weakly coupled Bose-Einstein condensates, *Phys. Rev. Lett.*, 96, 130404 (2006).
49. L. Pezzè, A. Smerzi, M.K. Oberthaler, R. Schmied and P. Treutlein, Quantum metrology with nonclassical states of atomic ensembles, *Rev. Mod. Phys.* 90, 035005 (2018).

Acknowledgements

Funded by the European Union (ERC, SUPERSOLIDS, n. 101055319) and by the QuantERA Programme that has received funding from the European Union’s Horizon 2020 research and innovation programme under Grant Agreements n. 731473 and n. 101017733, projects MENTA, SQUEIS and MAQS, with funding organisation Consiglio Nazionale delle Ricerche. **We acknowledge financial support from PNRR MUR project PE0000023-NQSTI financed by the European Union - Next Generation EU.** We gratefully acknowledge technical assistance from A. Barbini, A. Hajeb, F. Pardini, M. Tagliaferri and M. Voliani.

Contributions

N. A., G. B., A. F., C. G., L. T. and G. M. performed the experimental measurements. B. D., L. P., and A. S. developed the theoretical model and carried out the numerical simulations. N. A., G. B., B. D., and L. P. performed the experiment-theory comparison. N. A., G. B., B. D., L. P., A. S., A. F., M. F., C. G., M. I., L. T. and G. M contributed to the interpretation of the results and the writing of the paper.

Methods

Supersolid preparation

The experiment starts from a Bose-Einstein condensate (BEC) of ^{162}Dy atoms trapped in a harmonic potential created by two dipole traps crossing in the horizontal (x,y) plane⁵⁰. To tune the interaction parameter $\epsilon_{dd} = a_{dd}/a_s$, we control the contact scattering length a_s with magnetic Feshbach resonances, while the dipolar scattering length $a_{dd} = 130 a_0$ is fixed. The condensate is initially prepared at a magnetic field $B \approx 5.5$ G, corresponding to a scattering length of about $140 a_0$. The magnetic field is then slowly changed towards the critical values for the superfluid-supersolid phase transition, close to the set of Feshbach resonances around 5.1 G^{7,51}. We calibrate the magnetic field amplitude via radio-frequency spectroscopy before and after each experimental run. The magnetic field stability is about 0.5 mG, corresponding to a scattering length stability of about $0.25 a_0$. Since the overall systematic uncertainty in the absolute value of a_s is about $3 a_0$, corresponding to an uncertainty in ϵ_{dd} of about 4%, we identify a precise B -to- a_s conversion by comparing the experimental and numerical data for the critical ϵ_{dd} for the phase transition⁴⁰. The typical atom number in the supersolid is $N = (2.8 \pm 0.3) \times 10^4$. We expect thermal effects to be negligible in the Josephson dynamics, since the coupling energy $K(N_2 + N_3)$ is of the order of $k_B 100$ nK in the whole supersolid regime, and from measurements of the thermal fraction on the BEC side we get $T < 30$ nK⁷.

Excitation of the Josephson dynamics

The optical lattice used to excite the Josephson dynamics consists of two repulsive laser beams at 1064 nm that intersect at a small angle, providing a lattice period $d_L = (7.9 \pm 0.3)$ μm . The stability of the lattice position is better than 10% of its period over the duration of the experiment (see Extended Data Fig. 1a), and before each measurement we check the relative position of the lattice and the supersolid. Most of the noise in the excitation protocol comes from shot-to-shot fluctuations of the supersolid lattice due to the Goldstone mode (the position of a single cluster has a standard deviation of the order of 25% of the supersolid period, see below).

To calibrate the phase difference imprinted by the optical lattice, we switch on the lattice at fixed $U_{lat} = k_B 100$ nK for a variable pulse duration τ and we measure the imprinted phase difference $\varphi_0 = U_{lat} \tau / \hbar$ immediately after the pulse, see Extended Data Fig. 1b.

In the experiment, we detect clear Josephson oscillations only when the initial imprinted phase is 1 rad or larger. In this regime, we compare the experimental and numerical Josephson frequencies as a function of the amplitude of the oscillation (see Extended Data Fig. 2). We find a small reduction (about 15%) compared to the small-amplitude regime, which allows us to use the equation $\omega_j = \sqrt{4KN_{34}U}$ to extract the coupling energy K from the experimental Josephson frequencies. The need for large excitation amplitudes in the experiment can be explained by the presence of the Goldstone mode, which introduces an unavoidable noise on both Z and $\Delta\varphi$.

We checked that applying the same phase imprinting protocol to standard Bose-Einstein condensates does not produce any detectable Josephson oscillation, see Extended Data Fig. 3. This observation can be justified by the fact that the spatially-stationary excitations of

the condensate, the rotons, have a spatial period similar to the supersolid period d , so they cannot be excited by the optical lattice with $d_L \approx 2d$. The excitations with spatial period equal to d_L have instead a phonon/maxon character, they are not stationary in the harmonic trap and so they cannot produce spatially stable oscillations.

In general, this observation proves that the self-induced Josephson oscillations exist in the supersolid but not in the superfluid. We cannot make reliable experiments in the solid-like droplet-crystal phase, due to the exceedingly short lifetime of the experimental system in that regime, but the simulations show that the Josephson coupling becomes negligible and Josephson oscillations are absent.

Phase detection and analysis

To measure the phase difference between the two central clusters of the supersolid, $\Delta\varphi = \varphi_3 - \varphi_4$, we record the atomic distribution in the (x,y) plane by absorption imaging after 61 ms of free expansion. About 200 μs before releasing the atoms from the trapping potential, we increase the contact interaction strength by setting $a_s = 140 a_0$, thus minimising the relative effects of the long-range dipolar interaction on the expansion. We interpret the recorded distributions as the atomic density in momentum space, $\rho(k_x, k_y)$. In the supersolid regime, the momentum distribution shows an interference pattern due to the superposition of the expanding matter-waves of each cluster (see snapshots in Fig. 2a). We first integrate the two-dimensional distribution over k_y to obtain the one-dimensional momentum distribution $\rho(k_x)$. We then fit $\rho(k_x)$ with a double-slit model :

$$\rho(k_x) = G(k_x, k_0, \sigma) \left[1 + A_1 \cos^2(\pi(k_x - k_0)/k_r + \theta) \right]$$

where $G(k_x, k_0, \sigma)$ is a gaussian envelope of centre k_0 and width σ , while A_1, k_r and θ are the amplitude, period and phase of the modulation, respectively. Due to the $\cos^2(x)$ term in our fit function, the physical phase difference is given by $\Delta\varphi = 2\theta$.

Although the interference pattern is generated by six overlapping clusters, $\Delta\varphi$ can be extracted with a good approximation (within 20%) by the double-slit model due to the finite resolution of our imaging system in momentum space ($0.2 \mu\text{m}^{-1}$, $1/e$ Gaussian width) and to the lower weight of lateral clusters. This is experimentally confirmed by the measured imprinted phase φ_0 as a function of the pulse depth, shown in ExtendedData Fig. 1, which is in good agreement with the prediction for the phase difference between adjacent clusters, $U_{lat} \tau / \hbar$.

For each observation time t , we take $n = 20-30$ images. We then calculate the mean value of $\Delta\varphi$ using the circular mean, which is appropriate for a periodic quantity such as an angle:

$$\overline{\Delta\varphi} = \arg \left(\sum_{j=1}^n e^{i\Delta\varphi_j} \right)$$

where $\arg(x)$ indicates the argument of the complex number x and i is the imaginary unit. The corresponding error is given by the circular standard deviation **of the mean**⁵².

Imbalance detection and analysis

To measure the population imbalance between clusters, we image the supersolid in-situ in the (x,y) plane using an imaging system with a resolution of $2.5 \mu\text{m}$, smaller than the cluster spacing of $4 \mu\text{m}$. To avoid saturation effects due to the high density of the sample, we use

dispersive phase-contrast imaging⁵³ with an optical beam detuned by 5Γ from the 421 nm optical transition. From each experimental shot, we calculate the imbalance as follows. We integrate the column density along the y-direction (transverse to the modulation) obtaining 1D density profiles where we identify the two main peaks (snapshots in Fig. 2b). We then measure the populations $N_1 + N_2 + N_3$ and $N_4 + N_5 + N_6$ integrating the signal to the left and to the right of the minimum between the clusters, respectively. We then compute the observable $Z = (N_6 + N_5 + N_4 - N_3 - N_2 - N_1)/N$.

Due to the limited optical resolution, we can only clearly resolve the left and right clusters populations when the contrast of the density modulation is high enough, i.e. only at $\epsilon_{dd} = 1.444$. For lower ϵ_{dd} , we use an optical separation technique to increase the signal-to-noise ratio. We turn on the optical lattice used for the excitation 5 ms before image acquisition. This causes the main clusters to move away, falling into the minima of the optical potential and increasing their distance (snapshots in Fig. 2a and in Extended Data Fig. 4). Although our lattice does not have the optimal spatial phase to separate the clusters, since it has a maximum at the position of one cluster, we checked with numerical simulations that the only effect on the imbalance Z is the addition of a constant offset, thus not changing the oscillation frequency (see Extended Data Fig. 4). Experimentally, we checked that the Josephson frequencies measured with and without the optical separation, are consistent within one standard deviation (see filled and empty blue points at $\epsilon_{dd}=1.444$ in Fig.4). At lower ϵ_{dd} , very close to the phase transition, the contrast is too low, so we rely only on phase measurements.

Experimental measurement of the superfluid fraction from the Josephson frequency

To measure the superfluid fraction in the whole supersolid regime, reported in Fig.4, we employ the Josephson frequency ω_j extracted from phase oscillations. We use the formula

$f_s = \frac{\omega_j^2/(4N_{34}U)}{\hbar^2/(2md^2)}$. The period d of the supersolid lattice is measured with in situ imaging, obtaining $d = 3.7 \pm 0.1 \mu\text{m}$. The quantity $N_{34}U$ is taken from the numerical simulations.

Since the experimental oscillations are not in the small amplitude limit, the frequencies are underestimated by about 15% (see Fig. Extended Data 2). The upper error bar for f_s in Fig. 4 includes accordingly a 15% uncertainty. For the experimental configurations where we measure both Z and $\Delta\varphi$, we also checked that U extracted from Eq. 5b is in agreement with the simulations.

Discussion of the Leggett model

Leggett derived the upper bound for the superfluid fraction f_s^u in the case of a 1D system rotating in an annulus with radius R , for which the moment of inertia is $I = (1 - f_s)I_c$, with I_c the classical moment of inertia¹⁶. To find the phase profile $\varphi(x)$ that minimises the kinetic energy for a fixed number density $n(x)$, one has to work in the frame corotating with the annulus, in which the external potential is time-independent. In this frame, the rotation imposes a phase twist between neighbouring clusters, proportional to the angular velocity Ω of the annulus, $\Delta\varphi = \varphi(d) - \varphi(0) = m\Omega Rd/\hbar$. The result of the energy minimisation is

$\varphi(x) = \Delta\varphi \int_0^x dx' n(x')^{-1} / \int_0^d dx' n(x')^{-1}$ and the corresponding kinetic energy cost is

$E_{kin}(\Delta\varphi) = N\hbar^2/(2md^2)f_s^u\Delta\varphi^2$, where f_s^u is the upper bound of Eq. (3). The lower bound f_s^l , instead, is found starting from the 3D kinetic energy, which includes also the derivatives along the transverse directions y and z ³⁴. It reads $f_s^l = \int dydz \left(1/d \int_0^d dx/\bar{n}(x,y,z)\right)^{-1}$, where $\bar{n}(x,y,z)$ is the normalised 3D density. From the expression of the energy, we see that the superfluid fraction has the role of an elastic constant for the phase deformation. The density and phase profiles sketched in Fig. 1b, and the corresponding energy density $\hbar^2/(2m)n(x)|\nabla\varphi(x)|^2$, are for an hypothetical homogeneous supersolid lattice with $f_s = 0.20$.

In the Josephson case, the phase twist is externally applied with an odd parity, to induce Josephson oscillations between neighbouring sites. The energy minimisation on the single cell gives the same result as before, since it is insensitive to the sign of the phase twist. In the sketch of Fig.1c we build the odd phase profile by changing sign from cell to cell to $\varphi(x)$ of Fig. 1b. **We note that in a linear system like the one employed in the experiment, the superfluid fraction measured from the Josephson dynamics is not affected by radial effects, which are instead relevant in the case of rotating systems³¹. Indeed, the superfluid fraction extracted from a measurement of the moment of inertia, $I = (1 - f_s)I_c$, would also take into account the additional contribution given by the reduced inertia of the superfluid clusters composing the system, which rotate around their centers of mass. Leggett's upper bound is instead derived in the limit of an infinite radius of the annulus, where such radial effects can be neglected¹⁶.**

Goldstone Mode

In a harmonic trap, the Goldstone mode energy $\hbar\omega_G$ is finite but much smaller than $\hbar\omega_x$, since the supersolid can rearrange its density to minimise the centre of mass displacement. The resulting dynamics is an oscillation of the lattice position, imbalance and relative phase. Due to its low frequency, the Goldstone mode is thermally activated. Similarly to previous works²⁶, we detect the Goldstone excitation as fluctuations in the lattice position that keep the centre of mass fixed (see Extended Data Fig. 5). We prepare the supersolid with three main clusters and, without any additional manipulation, we probe the in-situ density. We detect fluctuations in the cluster positions, with standard deviation $\sigma_{clusters} \sim 1 \mu\text{m}$, much larger than the centre of mass fluctuations, $\sigma_{com} \sim 0.4 \mu\text{m}$. The Goldstone mode also introduces some noise in Z and $\Delta\varphi$ during the dynamics, which we estimate to be about 20% of the Josephson amplitude, for both observables.

The frequency of the Goldstone mode can be observed in numerical simulations at $T=0$ by setting an initial $Z_0 > 0$ together with a small displacement of the weak link position, $x_0 \neq 0$, see Extended Data Fig. 5. In the time evolution of Z , we find a very low frequency oscillation, $\omega_G = 2 \text{ Pi } (3.56 \pm 0.08) \text{ Hz}$, on top of the Josephson dynamics, $\omega_J = 2 \text{ Pi } (23.85 \pm 0.03) \text{ Hz}$. We find similar values for $\Delta\varphi$. The weak link position oscillates at the same low frequency ω_G .

Numerical simulations

To simulate the dynamics of our system we numerically integrate the extended Gross-Pitaevskii equation (eGPE):

$$i\hbar \frac{\partial \psi(\bar{r}, t)}{\partial t} = \left[-\frac{\hbar^2}{2m} \nabla^2 + V_{h.o.}(\bar{r}) + g |\psi(\bar{r}, t)|^2 + \int d\bar{r}' V_{dd}(|\bar{r} - \bar{r}'|) |\psi(\bar{r}', t)|^2 + \gamma(\epsilon_{dd}) |\psi(\bar{r}, t)|^3 \right] \psi(\bar{r}, t)$$

where $V_{h.o.}(\bar{r}) = \frac{1}{2}m(\omega_x^2 x^2 + \omega_y^2 y^2 + \omega_z^2 z^2)$ is the harmonic external potential, $g = \frac{4\pi\hbar^2 a_s}{m}$

is the contact interaction parameter, $V_{dd}(r) = \frac{C_{dd}}{4\pi} \frac{1-3\cos^2\theta}{r^3}$ is the dipolar interaction, with θ the

angle between r and \hat{z} and $C_{dd} = 3\epsilon_{dd}g$. The last term is the Lee-Huang-Yang energy of quantum fluctuations⁵⁴. Josephson dynamics was induced either by an antisymmetric phase imbalance imprinted with a sinusoidal potential as in the experiment, or by an initial antisymmetric population imbalance. Both methods excite the same Josephson normal mode. Atom number and phase for each cluster are calculated at each time step by determining the position of the density minima between the clusters, eliminating their slow and weak oscillations.

The superfluid fraction in Fig. 4a (green dots) is obtained by calculating the coupling energy K in the limit of small initial imbalance ($Z(0) \approx 0.01$), finding values in the range $K \sim k_B(0.1 - 0.01)$ nK. From Eq. (5b) we find $N_{34} U \sim k_B(5 - 7)$ nK, slowly varying with ϵ_{dd} . The ratio $N_{34} U / (2K)$ is always larger than 25.

Six-mode Josephson model

We employ a set of six-mode Josephson equations with interaction parameters U_j , with $j=1, \dots, 6$ labelling the clusters, five coupling parameters between adjacent clusters K_{jj+1} , and energy offsets E_0 and E_1 , for the opposite side clusters 1 and 6, and 2 and 5, due to the harmonic trap. We indicate as $K = K_{34}$ and $U = U_3 = U_4$ the coupling and interaction energy, respectively, in two central clusters. The symmetry of the system further allows us to equalise the two side coupling $K' = K_{23} = K_{45}$ and $K'' = K_{12} = K_{56}$, the two side interactions $U' = U_2 = U_5$ and $U'' = U_1 = U_6$ (see Fig. 2a). We thus have a system of six equations for the time evolution of the populations N_j and five phase differences $\varphi_{ij} = \varphi_i - \varphi_j$:

$$\begin{aligned} \dot{N}_1 &= -2K'' \sqrt{N_2 N_1} \sin(\varphi_{21}) \\ \dot{N}_2 &= 2K'' \sqrt{N_2 N_1} \sin(\varphi_{21}) - 2K' \sqrt{N_3 N_2} \sin(\varphi_{32}) \\ \dot{N}_3 &= 2K' \sqrt{N_3 N_2} \sin(\varphi_{32}) - 2K \sqrt{N_4 N_3} \sin(\varphi_{43}) \\ \dot{N}_4 &= 2K \sqrt{N_4 N_3} \sin(\varphi_{43}) - 2K' \sqrt{N_5 N_4} \sin(\varphi_{54}) \\ \dot{N}_5 &= 2K' \sqrt{N_5 N_4} \sin(\varphi_{54}) - 2K'' \sqrt{N_6 N_5} \sin(\varphi_{65}) \\ \dot{N}_6 &= 2K'' \sqrt{N_6 N_5} \sin(\varphi_{65}) \end{aligned} \quad (5a)$$

$$\begin{aligned}
\dot{\varphi}_{21} &= E_1 + U''N_1 - U'N_2 \\
\dot{\varphi}_{32} &= E_0 + U'N_2 - UN_3 \\
\dot{\varphi}_{43} &= U(N_3 - N_4) \\
\dot{\varphi}_{54} &= -E_0 + UN_4 - U'N_5 \\
\dot{\varphi}_{65} &= -E_1 + U'N_5 - U''N_6
\end{aligned} \tag{5b}$$

where we have considered the case $(N_4 + N_3)U/(2K) \gg 1$ so that we have neglected the tunnelling terms in the evolution of the phases.

In the following we further consider small amplitude oscillations such that we can replace $\sqrt{N_i N_j} \approx \sqrt{N_i(0)N_j(0)}$ and $\sin(\varphi_{ij}) \approx \varphi_{ij}$ into Eq. (5a), where $N_j(0)$ is the initial population of the j th cluster at time $t = 0$. For symmetry reasons, we have $N_1(0) \approx N_6(0)$, $N_2(0) \approx N_5(0)$, and $N_3(0) \approx N_4(0)$. Even in the linear regime, the time evolution of populations and phases predicted by Eqs. 5(a-b) shows multiple frequencies. Harmonic single-frequency oscillations with a $\pi/2$ phase shift between populations and relative phases are observed under the two conditions

$$\begin{aligned}
\frac{U''}{U'} &= 1 + \frac{K'}{K''} \sqrt{\frac{N_3(0)}{N_1(0)}}, \tag{6} \\
\frac{U'}{U} &= \frac{1 + \frac{K}{K'} \sqrt{\frac{N_3(0)}{N_2(0)}}}{1 + \frac{K''}{K'} \sqrt{\frac{N_1(0)}{N_3(0)}}}.
\end{aligned}$$

In particular, under these conditions, we have

$$\dot{N}_3 - \dot{N}_4 = \alpha(\dot{N}_6 - \dot{N}_1 + \dot{N}_5 - \dot{N}_2), \tag{7}$$

where $\alpha = 1/(U/U' - U/U'')$. The corresponding Josephson oscillation frequency is

$$\omega_J^2 = 2KU[N_3(0) + N_4(0)]\alpha/(\alpha-1). \tag{8}$$

To evaluate the parameters in the above equations and verify Eq. (6), we insert into Eqs. (5a) and (5b) the numerical results for $N_j(t)$ and $\varphi_{ij}(t)$ obtained from GPE simulations. A comparison between GPEs oscillations and the six-mode model is shown in Extended Data Fig. 6. First, the GPE ground state gives $N_3(0) = N_4(0) \approx N/4$, while the population of the lateral clusters depends on ε_{dd} . In particular, outer clusters $N_1(0) = N_6(0)$ decrease, while $N_2(0) = N_5(0)$ increase as ε_{dd} increases. The parameters U and K of the central clusters are extracted from Eqs. 4(a-b) in the main text. The other parameters U' , U'' , K' and K'' are extracted from fits using Eqs. 5(a-b). Overall, we obtain that the interactions parameters are $U/U' \approx 1$, $U/U'' \approx 1/2$ within fluctuations of about 10% for different values of ε_{dd} . On the

other hand, the coupling ratio $K/K' \approx 0.6$ is constant, while $K/K'' \approx 0.7$ on the BEC side and decreases with ϵ_{dd} , as do the initial external populations $N_1(0) = N_6(0)$. We thus find that Eq. (7) is fulfilled and $\alpha=2$. For this value of α , Eq. (8) gives $\omega_j^2=4KU[N_3(0)+N_4(0)]$, in agreement with the main text.

Taking into account Eq. (7) and the symmetry condition $N_3(0) = N_4(0)$, we find $N_3 - N_4 = \alpha(N_6 - N_1 + N_5 - N_2)$ at each time. We thus have $Z = (\alpha - 1)/\alpha \Delta N/N$, with $\Delta N = N_3 - N_4$. This reduces to $\Delta N = 2NZ$ for $\alpha = 2$. Using Eqs. (5a), we have $\dot{Z} = -4K\sqrt{N_4(0)N_3(0)}/N \sin(\Delta\varphi)$, with $\Delta\varphi = \varphi_{43}$. We can write $2\sqrt{N_3(0)N_4(0)} = N_3(0) + N_4(0) = N_{34}$ and get [deleted formula: $\dot{Z} = -4KN_{34}/N \sin(\Delta\varphi)$] $\dot{Z} = -2KN_{34}/N \sin(\Delta\varphi)$. The evolution of the phase difference $\Delta\varphi = U(N_3 - N_4)$, see Eq. (5b), rewrites as $\dot{\Delta\varphi} = U\Delta N = 2NUZ$.

It is interesting to take the limit of an infinite array of equal junctions, each one characterised by the same parameters of our central cell. Eqs. 5(a) are all equivalent, and due to the symmetry of the array, $N_i(0) = N_j(0)$ and $\varphi_{i+1,i} = -\varphi_{i,i-1} \forall i,j$. We then get $\dot{\Delta N} = \dot{N}_i - \dot{N}_{i+1} = -4KN_{i,i+1} \sin(\varphi_{i,i+1})$, with $N_{i,i+1} = N_i(0) + N_{i+1}(0)$, equivalent to Eq. (4a) in the main text. Eq. (4b) for the phase evolution applies in the infinite case as well.

Data availability

All data of the figures in the manuscript and Methods are available in a Zenodo repository <https://doi.org/10.5281/zenodo.10045059>

Code availability

The codes that support the findings of this paper are available from the corresponding author upon reasonable request.

Bibliography Methods

50. E. Lucioni L. Tanzi, A. Fregosi, J. Catani, S. Gozzini, M. Inguscio, A. Fioretti, C. Gabbanini, and G. Modugno, Dysprosium dipolar Bose-Einstein condensate with broad Feshbach resonances, Phys. Rev. A 97, 060701(R) (2018).

51. F. Bottcher, M. Wenzel, J. Schmidt, M. Guo, T. Langen, I. Ferrier-Barbut, T. Pfau, R. Bombín, J. Sánchez-Baena, J. Boronat, and F. Mazzanti, Dilute dipolar quantum droplets beyond the extended Gross-Pitaevskii equation, Phys. Rev. Res. 1, 033088 (2019).

52. N. I. Fisher, Statistical analysis of circular data (Cambridge Univ. Press, 1993).

53. H. Kadau, M. Schmitt, M. Wenzel, C. Wink, T. Maier, I. Ferrier-Barbut and T. Pfau, Observing the Rosensweig instability of a quantum ferrofluid, Nature 530, 194–197 (2016).

54. R. A. P. Lima and A. Pelster, Quantum fluctuations in dipolar Bose gases, Phys. Rev. A 84, 041604(R) (2011).

Extended data

Extended Data Fig. 1 - (a) Stability of the optical lattice. Black dots are the relative positions of the density peaks of a BEC loaded into the optical lattice with respect to the average centre of mass, for 45 different measurements. The standard deviation of fluctuations for each lattice position is $\sigma_{lattice} \sim 0.35 \mu\text{m}$. (b) Calibration of the initial phase difference imprinted on the two main clusters as a function of the lattice pulse duration. Red dots are experimental data obtained by imprinting the optical lattice potential for different pulse durations. **Error bars are the standard deviation of the mean for a typical sample of 15-20 experimental data points.** The dashed line is the theoretical prediction ($U\tau/\hbar$) with a lattice depth $U = 100 \text{ nK}$.

Extended Data Fig. 2 - Josephson frequency as a function of the phase amplitude, theory (black points) and experiment (red points). The vertical error bars are extracted from the sinusoidal fit of the Josephson oscillations. The horizontal error bars are the standard deviation of the mean of the phase difference detected at $t=0$, after the phase imprinting. For typical experimental amplitudes $\pi/2$, we observe a frequency reduction of about 15% compared to the small excitation regime. The highlighted region is the relevant one for our experiment.

Extended Data Fig. 3 - Absence of Josephson oscillations for a standard Bose-Einstein condensate. Time evolution of the population imbalance after the phase imprinting for a supersolid (blue) and an unmodulated BEC (orange), extracted using the optical separation. The dashed lines are sinusoidal fits with confidence bands (for 1σ) in shaded colour. The fitted oscillation amplitude of the BEC is compatible with zero. The error bars are the standard deviation of the mean of about 15-20 data points.

Extended Data Fig. 4 - Analysis of the optical separation technique. (a) Experimental in-situ images of two balanced ($Z = 0$) supersolids without any manipulation (left) and using the optical separation (right). (b) Numerical simulation of the dynamics of three different supersolids with initial population imbalance $Z_0 = 5\%$ (green), $Z_0 = 0\%$ (red) and $Z_0 = -5\%$ (blue), during the optical separation. Top row: imbalance Z . Bottom row: distance between the central clusters.

Extended Data Fig.5 - Evidence of the Goldstone mode in experimental and numerical data. (a) Left panels: fluctuations of the clusters positions (black and green points) and centre of mass (pink points) of the supersolid for about 100 experimental shots. Right panel: histograms of the right-cluster (green) and of the centre of mass (pink) positions. (b) Simulation of the Josephson dynamics coupled to the Goldstone oscillation. The

position of the density minimum x_0 (top row) shows a clear oscillation at the Goldstone frequency $\omega_G \ll \omega_J$. This lower frequency appears also in Z and $\Delta\varphi$ (bottom row) on top of the standard Josephson dynamics.

Extended Data Fig. 6 - Six-mode model and numerical simulations. (a) Sketch of the inhomogeneous system with six clusters with interaction energies U , U' and U'' , with coupling energies K , K' and K'' . The modes in the sketch are not on scale (compare with the simulation of our system in Fig. 2a in the main text). (b-c) Comparison between the time evolution calculated from the GPE simulations (thick lines) and from the six-mode model (dot-dashed lines) for Z (b) and $\Delta\varphi$ (c). (d) Relative currents between the central and lateral clusters appearing in Eq.(7), from GPE simulations. The solid line is $(\dot{N}_3 - \dot{N}_4)/2$ while the dashed line is $(\dot{N}_6 + \dot{N}_5 - \dot{N}_2 - \dot{N}_1)$.

Reviewer Reports on the First Revision:

Referees' comments:

Referee #1 (Remarks to the Author):

I am fine with the changes and support publication.

Referee #3 (Remarks to the Author):

I thank the authors for their detailed response and in my opinion the paper can now be published.